# IDEAL: Influence-Driven Selective Annotations Empower In-Context Learners in Large Language Models

**Shaokun Zhang**[1][*]    **Xiaobo Xia**[2][*][†]    **Zhaoqing Wang**[2]    **Ling-Hao Chen**[3]    **Jiale Liu**[4]
**Qingyun Wu**[1][†]    **Tongliang Liu**[2]
[1]Pennsylvania State University    [2]The University of Sydney
[3]Tsinghua University    [4]Xidian University
shaokun.zhang@psu.edu    xiaoboxia.uni@gmail.com

## Abstract

In-context learning is a promising paradigm that utilizes in-context examples as prompts for the predictions of large language models. These prompts are crucial for achieving strong performance. However, since the prompts need to be sampled from a large volume of annotated examples, finding the right prompt may result in high annotation costs. To address this challenge, this paper introduces an influence-driven selective annotation method that aims to minimize annotation costs while improving the quality of in-context examples. The essence of our method is to select a pivotal subset from a large-scale unlabeled data pool to annotate for the subsequent sampling of prompts. Specifically, a directed graph is first constructed to represent unlabeled data. Afterward, the influence of candidate unlabeled subsets is quantified with a diffusion process. A simple yet effective greedy algorithm for unlabeled data selection is lastly introduced. It iteratively selects the data if it provides a maximum marginal gain with respect to quantified influence. Compared with previous efforts on selective annotations, our influence-driven method works in an end-to-end manner, avoids an intractable explicit balance between data diversity and representativeness, and enjoys theoretical support. Experiments confirm the superiority of the proposed method on various benchmarks, achieving better performance under lower time consumption during subset selection. The project page is available at https://skzhang1.github.io/IDEAL/.

## 1 Introduction

In-context learning (ICL) entails presenting a small set of examples with demonstrations as prompts (called in-context examples) to large language models (LLMs), before making predictions on test inputs (Wei et al., 2022a; Min et al., 2022; Akyürek et al., 2023). This emerging few-shot learning paradigm is an appealing alternative to supervised fine-tuning as it can avoid heavy parameter updates of language models while improving accuracy (Liu et al., 2021; Yoo et al., 2022).

Recent studies indicate that obtaining prompts from a vast collection of annotated examples is crucial to achieving strong performance (Rubin et al., 2022). Notably, these studies have illuminated the substantial performance improvements when retrieving analogous examples (under specific embedding criteria) as in-context examples tailored for each individual test input. Since different test scenarios need distinct in-context examples, and each of them is equipped with its pertinent annotations, the necessity of a large volume of annotated examples is emphasized (Su et al., 2023). However, obtaining large-scale annotated examples for ICL requires substantial manpower and financial resources (Baldridge & Osborne, 2004; Engelson & Dagan, 1996; Snow et al., 2008). This is because humans not only need to annotate the true label for each example but also need to provide the example demonstration in the annotation process (Wei et al., 2022b).

---

[*]Equal contributions.
[†]Corresponding authors.

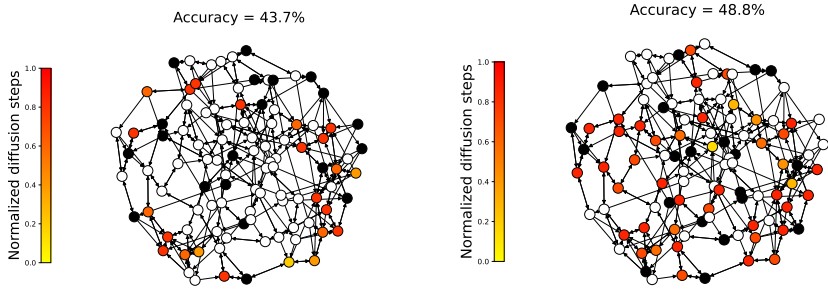

(a) Low-influence subset in unlabeled data.  (b) High-influence subset in unlabeled data.

Figure 1: Visualization of the information diffusion process (Goldenberg et al., 2001) of two subsets with equal sizes. Experiments are conducted using the SST-5 training set (Socher et al., 2013). To avoid the denseness, we randomly sample 100 examples in total. In this visualization, black nodes present the initial subset without information diffusion. White nodes correspond to the examples that are not influenced by diffusion. For other nodes, darker nodes represent earlier influenced examples. We can observe that the subset with high influence (b) can achieve better performance by influencing a larger group of examples in the unlabeled data pool compared to the subset with low influence (a).

To reduce the annotation cost, the previous effort Vote-$k$ (Su et al., 2023) made attempts by proposing to select a *diverse* and *representative* subset from a large-scale unlabeled data pool to annotate. Particularly, Vote-$k$ initially selects a small portion of data for diversity and annotates them manually. Then, these annotated data act as prompts for predictions on all other unlabeled data, and choose the remaining ones that need to be annotated, based on diverse confidence scores. However, despite its strong performance in empirical evaluations, Vote-$k$ is still unsatisfactory in practice. We detail the issues from three aspects. (1) The data selection procedure of Vote-$k$ is not end-to-end. This results in inconvenience, increased processing complexity, and added inference costs due to the predictions on unlabeled data. (2) Diversity and representativeness need to be balanced carefully (Su et al., 2023). Highlighting diversity in data selection is crucial for comprehensive coverage, but may sacrifice representativeness by overlooking exemplary data. Besides, the excessive emphasis on diversity of Vote-$k$ causes the selection of outliers (see evidence in Appendix C.2). (3) Vote-$k$ lacks theoretical guarantees, making it challenging to assess the algorithm's reliability in realistic tasks and constraining its practical utility.

In this paper, to minimize annotation costs for ICL and address the issues of existing work, an innovative data selection method is introduced, where we utilize influence-driven selective annotations to empower in-context learners (**IDEAL**). In essence, IDEAL aims to identify a subset of data that acts as a proxy and closely approximates the vast unlabeled dataset. Once annotated, these selected data can be considered a viable substitute for the large annotated examples in subsequent ICL tasks. In further detail, our method works in an unsupervised and *end-to-end* manner. We first construct a *directed graph*, where its vertices represent unlabeled data and its edges bridge different data based on their similarities. Inspired by influence maximization that aims to select a vertex set at key positions in social graphs (Li et al., 2018), we then propose to quantify the influence of each candidate unlabeled subset in our constructed graph, through a classic independent-cascade diffusion model illustrated in Figure 2. To find the subset with high influence, a simple greedy algorithm for unlabeled data selection is introduced. The algorithm does not need a delicate trade-off between diversity and representativeness. Instead, it iteratively selects a vertex if it provides a maximum marginal gain to the influence metric, until the selection is completed based on the annotation budget.

Theoretically, under the influence-driven selective paradigm, we provide the lower bound for the subset influence selected by our method, demonstrating it is at least as large as a certain proportion of the influence of the optimal solution. Empirically, we conduct comprehensive experiments over 9 datasets across diverse tasks (covering classification, commonsense reasoning, dialogue, and text/code generation). Various LLMs and prompt retrieval technologies are included in evaluations. Experimental results demonstrate that our IDEAL can achieve better performance than Vote-$k$ in 17 out of 18 cases in the experiments, with only 13% time consumption during subset selection. This creates a strong baseline of selective annotations for follow-up research. Source codes have been attached for the reproducibility of results.

## 2 METHODOLOGY

In this section, to reduce the annotation cost of ICL, a framework of influence-driven selective annotations is formulated. We discuss how examples should be selected to annotate, leading to better in-context learners for LLMs.

### 2.1 PROBLEM SETUP

We begin by defining notations and setting up the research problem. Specifically, LLMs perform in-context learning tasks based on a task-specific prompt $\mathbf{Z} = [\mathbf{z}_1, \ldots, \mathbf{z}_c]$, where each $\mathbf{z}_i$ represents one example $(\mathbf{x}_i, y_i)$ consisting of the instance $\mathbf{x}_i$ and label $y_i$, with $c$ examples in total. LLMs generate the prediction for one test input $\mathbf{x}_{\text{test}}$ conditioned on the prompt $\mathbf{Z}$ followed by $\mathbf{x}_{\text{test}}$, i.e., $y_{\text{test}} = \arg\max_{y \in \mathcal{C}} P(y|\mathbf{Z}, \mathbf{x}_{\text{test}})$, where $\mathcal{C}$ denotes the label space. As each prompt needs distinct annotations, the importance of having a substantial number of annotated examples is stressed, resulting in huge annotation costs. This motivates us to investigate selective annotations.

Given a pool of unlabeled instances $\mathcal{D}_{\text{u}} = \{\mathbf{x}_i\}_{i=1}^n$, where $n$ is the number of unlabeled instances, the aim of selective annotations is to select a subset $\mathcal{S}_{\text{u}} \subset \mathcal{D}_{\text{u}}$ to make manual annotations, such that performing ICL using prompts retrieved from the selected subset can yield good performance on an unseen test set $\mathcal{D}_{\text{test}}$. The size of $\mathcal{S}_{\text{u}}$ is controlled by the annotation budget $m$, i.e., $|\mathcal{S}_{\text{u}}| = m$.

### 2.2 INFLUENCE-DRIVEN SELECTIVE ANNOTATIONS

**Overview.** For selective annotations in ICL, we need to identify a subset that approximates vast unlabeled data. Therefore, quantifying the coverage of each candidate subset is critical. To achieve this, we construct a directed graph using the embeddings of unlabeled data and portray their relationships using the edges in the graph. We then quantify the influence of each candidate subset in the constructed graph. An information diffusion model is used for this purpose. Through the information diffusion model to quantifying the influence of each candidate subset, we avoid the delicate trade-off between diversity and representativeness. After the quantification, we can search the subset with maximum influence, which most closely approximates the unlabeled data. Below we detail the above procedure step by step.

**Constructing the directed graph.** We first compute a vector embedding for each unlabeled instance using Sentence-BERT (Reimers & Gurevych, 2019)[1]. The obtained embeddings are employed to build a directed graph $\mathcal{G} = (\mathbf{V}, \mathbf{E}, \mathbf{P})$, where the vertices $\mathbf{V} = \{\mathbf{v}_i\}_{i=1}^n$ represent the embeddings of the unlabeled instances, $\mathbf{E}$ denotes the set of edges in the graph, and $\mathbf{P}$ denotes the set of weights assigned to edges. In more detail, for each vertex $\mathbf{v} \in \mathbf{V}$, we connect it to its $k$ nearest successors[2] in terms of the cosine similarity between the embeddings and then get $\mathbf{E}$. For the edge $(\mathbf{v}, \mathbf{u}) \in \mathbf{E}$ that connects $\mathbf{v}$ and its successor $\mathbf{u}$, we assign the weight $p(\mathbf{v}, \mathbf{u}) = \cos(\mathbf{v}, \mathbf{u})/\sum_{\mathbf{z} \in \mathcal{N}(\mathbf{v}, k)} \cos(\mathbf{v}, \mathbf{z})$ with $p \in \mathbf{P}$, where $\mathcal{N}(\mathbf{v}, k)$ represents the set including $k$ nearest successors of $\mathbf{v}$, and $\cos(\cdot, \cdot)$ is a function that calculates the cosine similarity of two embeddings. The constructed graph depicts the relationships between unlabeled examples in terms of the embedding similarity.

**Quantifying subset influence.** Here we propose to quantify each candidate subset within the constructed graph, which is detailed in Algorithm 1. Specifically, given the constructed graph $\mathcal{G}$ and a candidate subset $\mathcal{S}$, the quantification algorithm simulates the *progression of information diffusion* originating from $\mathcal{S}$. The number of influenced vertices can be considered as a measure of the influence of the candidate subset. In other words, the subset that influences more vertices within the graph can provide a better approximation of the vast unlabeled data. The diffusion process unfolds discretely, progressing through multiple steps. At the beginning, the subset $\mathcal{S}$ is activated. Then at each step, each vertex $\mathbf{v}$ activates its successors that remained inactive in the last step with a probability defined by $p(\mathbf{v}, \mathbf{u})$. The activation can be conceptualized as a coin toss where the outcome is determined by the head probability $p(\mathbf{v}, \mathbf{u})$. If the result is the head, the vertex $\mathbf{v}$ becomes activated; otherwise, it remains inactive. Starting from $\mathcal{S}$, the diffusion terminates when no further vertex can be activated in the graph. Lastly, we quantify the influence of the set with the number of activated vertices, where a larger number corresponds to greater influence. In order to get a stable result, we

---

[1] https://huggingface.co/sentence-transformers/all-mpnet-base-v2.
[2] In graph theory (Harary, 2018), a vertex $\mathbf{u}$ is the successor of a vertex $\mathbf{v}$ if it is at the end of an outgoing directed edge $(\mathbf{v}, \mathbf{u})$.

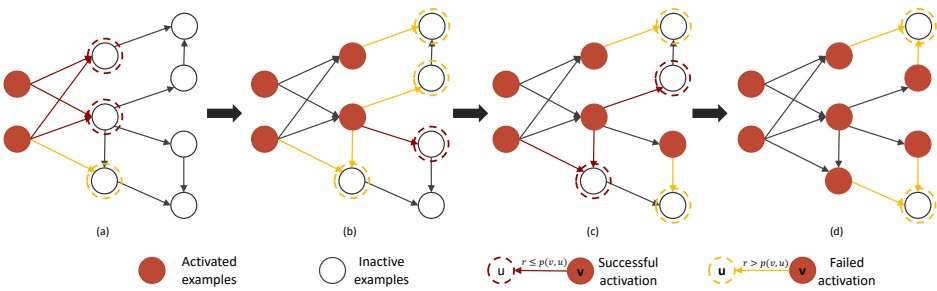

| | | | |
|---|---|---|---|
| ⬤ Activated examples | ◯ Inactive examples | (u) → (v) $r \le p(v,u)$ Successful activation | (u) → (v) $r > p(v,u)$ Failed activation |

Figure 2: The procedure aims to quantify the influence of each subset of in-context examples. In this procedure, we start with a subset of examples (the red points in (a)). Gradually, the successors of this subset are activated based on the weight $p$ and a random number $r$ sampled from 0 to 1. From (a) to (d). The influence of the subset is determined by the number of points that have been activated.

---

**Algorithm 1:** Subset influence quantification.

---

**Input** : Directed graph $\mathcal{G} = (\mathbf{V}, \mathbf{E}, \mathbf{P})$, subset $\mathcal{S}$.
**Output:** Number of influenced vertices by $\mathcal{S}$ in $\mathcal{G}$.
$\mathcal{S}_{\text{active}} \leftarrow \mathcal{S}, \mathcal{S}_{\text{new}} \leftarrow \emptyset, L = 0$;
**while** $\mathcal{S}_{\text{active}} \neq \emptyset$ **do**
  **for** *each node* $\mathbf{v}$ *in* $\mathcal{S}_{\text{active}}$ **do**
    **for** *each successor* $\mathbf{u}$ *of* $\mathbf{v}$ *in* $\mathcal{G}$ **do**
      **if** $\mathbf{u}$ *not in* $\mathcal{S}$ **then**
        Generate random number $\tau \in [0, 1]$;
        **if** $\tau \le p(\mathbf{v}, \mathbf{u})$ **then**
          $\mathcal{S} \leftarrow \mathcal{S} \cup \mathbf{u}; \mathcal{S}_{\text{new}} \leftarrow \mathcal{S}_{\text{new}} \cup \mathbf{u}$;

  $\mathcal{S}_{\text{active}} \leftarrow \mathcal{S}_{\text{new}}; L \leftarrow L + |\mathcal{S}_{\text{new}}|; \mathcal{S}_{\text{new}} \leftarrow \emptyset$;
**return** $L$.

---

**Algorithm 2:** Searching the subset with maximum influence.

---

**Input:** The directed graph $\mathcal{G} = (\mathbf{V}, \mathbf{E}, \mathbf{P})$, the annotation budget $m$.
**Result:** The set $\mathcal{S}_{\text{u}}$ that includes $m$ examples to annotate.
**Initialize** $\mathcal{S}_0 \leftarrow \emptyset, t = 0$;
**while** $t < m$ **do**
  $\mathbf{v}_t \leftarrow \arg\max_{\mathbf{v} \in \mathbf{V} \setminus \mathcal{S}_t} f_{\mathcal{G}}(\mathcal{S}_t \cup \{\mathbf{v}\})$;
  $\mathcal{S}_{t+1} \leftarrow \mathcal{S}_t \cup \mathbf{v}_t$;
  $t \leftarrow t + 1$;
**Obtain** $\mathcal{S}_{\text{u}}$ with $\mathcal{S}_m$ using the correspondence between embeddings and instances;
**return** $\mathcal{S}_{\text{u}}$.

---

repeat this process ten times and take the average influence. To help understand the procedure of Algorithm 1, we provide an illustration as shown in Figure 2. For convenience, we express Algorithm 1 as an influence function $f_{\mathcal{G}}(\mathcal{S})$ for the graph $\mathcal{G}$ that takes example set $\mathcal{S}$ as inputs, and returns the number of activated vertices $L$.

**Searching the subset with maximum influence.** We exploit a simple yet effective greedy algorithm (Kempe et al., 2003) to search the subset with maximum influence, which is illustrated in Algorithm 2. Specifically, the algorithm is initialized with an empty set, and iteratively involves an instance if it can provide the maximum marginal gain to the influence function. The search algorithm terminates when the selected subset meets the annotation budget. Finally, we achieve the set $\mathcal{S}_{\text{u}}$ that includes $m$ examples to annotate, using the correspondence between embeddings and instances. It is worth mentioning that this searching process aims to maximize the influence of the whole selected subset rather than considering each example separately. This is because combining all the individual high-impact examples together does not necessarily achieve the highest-impact subset.

## 2.3 PROMPT RETRIEVAL

After the above influence-driven selective annotations, the subset $\mathcal{S}_u$ is achieved. By making manual annotations on $\mathcal{S}_u$, a set of annotated examples is obtained. We can then retrieve examples from the annotated set as in-context examples for each test input. Following previous studies (Liu et al., 2021; Su et al., 2023), we will calculate embeddings for all annotated examples using Sentence-BERT (Reimers & Gurevych, 2019) and identify the most similar instances to each test input based on the cosine similarity. Notice that, the proposed method is agnostic to prompt retrieval methods. As demonstrated in §4.3.3, our method can be combined with any other prompt retrieval technologies. Better prompt retrieval technologies can further boost final performance.

## 3 THEORETICAL ANALYSIS

In this section, we perform theoretical analysis on the influence of the subset searched by our algorithm and provide the corresponding lower bound. For any constructed graph $\mathcal{G}$, we exploit $\psi_{\mathbf{v}}(\mathcal{S})$ to denote the influence improvement of the subset $\mathcal{S}$ after adding $\mathbf{v}$ into $\mathcal{S}$, i.e., $\psi_{\mathbf{v}}(\mathcal{S}) = f_{\mathcal{G}}(\mathcal{S} \cup \mathbf{v}) - f_{\mathcal{G}}(\mathcal{S})$. For convenience, we use $\psi_t = f_{\mathcal{G}}(\mathcal{S}_t) - f_{\mathcal{G}}(\mathcal{S}_{t-1})$ $(t \geq 1)$ to denote the incremental value of the influence function $f_{\mathcal{G}}$ after adding $\mathbf{v}_t$ into $\mathcal{S}_{t-1}$. Also, we employ $\mathcal{S}_m^*$ to represent the subset with the optimal influence value in the graph $\mathcal{G}$ with annotation budget $m$. Afterward, the optimal solution we expect to search in Algorithm 2 can be regarded as

$$\mathcal{S}_m^* = \arg\max_{\mathcal{S} \subset \mathbf{V}} f_{\mathcal{G}}(\mathcal{S}), \quad \text{s.t.} \quad |\mathcal{S}| = m. \tag{1}$$

In the following, we present the submodular condition to facilitate theoretical analysis of our method.

**Condition 1** (submodular condition). *In the problem of selective annotations, given any graph $\mathcal{G}$ constructed by our procedure, the influence function $f_{\mathcal{G}}$ is a submodular function which satisfies, for $\forall \mathbf{v} \in \mathbf{V}, \forall \mathcal{S}_a \subset \mathcal{S}_b \subset \mathbf{V}$,*

$$f_{\mathcal{G}}(\mathcal{S}_a \cup \mathbf{v}) - f_{\mathcal{G}}(\mathcal{S}_a) \geq f_{\mathcal{G}}(\mathcal{S}_b \cup \mathbf{v}) - f_{\mathcal{G}}(\mathcal{S}_b). \tag{2}$$

**Remark 1.** Intuitively speaking, given any graph $\mathcal{G}$, we say the influence function $f_{\mathcal{G}}$ satisfies the submodular condition if adding one data point to a smaller subset provides more influence than adding the same data point to a larger subset. In other words, it reflects the principle of diminishing returns: the marginal gain of including a data point in a set decreases as the size of the set increases. This condition can hold within the influence function (Li et al., 2019). Considering an extreme case, when subset $\mathcal{S} = \mathbf{V}$, the influence improvement of adding any data point to $\mathcal{S}$ will be zero.

**Proposition 1.** *In Algorithm 2, if the influence function $f_{\mathcal{G}}$ satisfies Condition 1, then for $f_{\mathcal{G}}(S_m^*)$,*

$$\forall t \in [0, m-1), f_{\mathcal{G}}(\mathcal{S}_m^*) \leq f_{\mathcal{G}}(\mathcal{S}_t) + m\psi_{t+1}. \tag{3}$$

**Remark 2.** Proposition 1 proposes an upper bound for $f_{\mathcal{G}}(\mathcal{S}_m^*)$ in the form of the influence $f_{\mathcal{G}}(\mathcal{S}_t)$ and its improvement at next step $t+1$, when Algorithm 2 is applied to selective annotations.

**Theorem 1.** *In Algorithm 2, if influence function $f_{\mathcal{G}}$ satisfies Condition 1, when the algorithm terminates at the step $m-1$, $f_{\mathcal{G}}(\mathcal{S}_m)$ has a lower bound:*

$$f_{\mathcal{G}}(\mathcal{S}_m) \geq (1 - (1 - 1/m)^m) f_{\mathcal{G}}(\mathcal{S}_m^*). \tag{4}$$

**Remark 3.** Theorem 1 provides an approximation guarantee for the influence of the selected subset returned by our method. The influence of the selected subset is at least as large as a certain proportion of the influence of the optimal solution, i.e., $1 - (1 - 1/m)^m$. With the annotation budget $m$ increases, this fraction gets closer to $1 - 1/e$.

For the proofs of Proposition 1 and Theorem 1, readers can refer to Appendix B.

## 4 EXPERIMENTS

In this section, we evaluate our method (IDEAL) on multiple datasets that have different categories of tasks. Experimental setups are first introduced (§4.1). We then demonstrate that the proposed method can find a better selective annotation subset in a more efficient way compared with baselines (§4.2). Moreover, we perform in-depth investigations to provide a better understanding of the superiority of the proposed method (§4.3). Finally, a case study is also provided to further evaluate the selected subset from our method in an automatic annotation scenario (§4.4).

### 4.1 EXPERIMENTAL SETUPS

**Datasets and tasks.** Following previous work (Su et al., 2023), we employ 9 datasets for the evaluations, which can be categorized into 4 different tasks, including classification, multi-choice, dialogue, and generation. The details of the datasets are provided in Appendix D.1. For each dataset, the original "train/dev/test" split from the Transformer library (Wolf et al., 2019) is utilized. We use test data for evaluation if they are available publicly (SST-5 (Socher et al., 2013), DBpedia(Lehmann et al., 2015), MWoZ (Budzianowski et al., 2018), and Xsum (Narayan et al., 2018)). Otherwise, we follow the same setting in (Su et al., 2023) and use the development set. We use accuracy as metric for all classifications and multiple choices tasks, joint accuracy (Budzianowski et al., 2018) for MWoZ, test suite accuracy (Zhong et al., 2020) for GeoQuery (Zelle & Mooney, 1996), and ROUGE-L (Lin, 2004) for Xsum.

**Models.** If not otherwise specified, we run all experiments on the GPT-J 6B model (Wang & Komatsuzaki, 2021) except the GeoQuery and MWoZ datasets where we use Text-devinci-002 (Chen et al., 2021). We also provide experiments on other models including GPT-Neo 2.7B (Black et al., 2021) and more advanced models GPT-3.5-Turbo (Openai, 2023) in §4.3.4. Our implementation is detailed in Appendix D.2.

**Baselines.** In the main experiments, we perform a comprehensive evaluation of our method that is compared with previous state-of-the-art selective annotation baselines, i.e., Vote-$k$ (Su et al., 2023) and random selection (abbreviated as "Random" below). Note that, in §4.3.2, we also compare our method with alternative methods that can select a coreset from large-scale unlabeled data on typical datasets. For the baseline Vote-$k$, we conduct experiments by running its official code[3].

### 4.2 MAIN RESULTS

| Method | Classification | | | | | Multi-Choice | Dialogue | Generation | |
|---|---|---|---|---|---|---|---|---|---|
| | MRPC | SST-5 | MNLI | DBpedia | RTE | HellaSwag | MWoZ | GeoQ | Xsum |
| 100 Random | 64.3 | 49.6 | 38.2 | 89.8 | 55.3 | 66.7 | 39.9 | 55.3 | 15.3 |
| 100 Vote-$k$ | 64.6 | 46.6 | 38.9 | 89.2 | 57.6 | 67.9 | 48.3 | **58.8** | 17.2 |
| 100 IDEAL | **66.4** | **51.4** | **41.0** | **90.6** | **58.9** | **68.6** | **52.2** | 58.2 | **19.9** |
| 18 Random | 57.4 | 42.9 | 37.8 | 85.2 | 57.9 | 66.0 | 37.0 | 47.5 | 13.6 |
| 18 Vote-$k$ | 61.1 | 41.7 | 39.1 | 89.9 | 58.2 | 66.5 | 37.7 | 50.9 | 15.2 |
| 18 IDEAL | **63.0** | **43.2** | **40.0** | **90.1** | **59.4** | **67.1** | **38.5** | **52.0** | **19.6** |

Table 1: The performance of our method and baselines on 9 different datasets with an annotation budget of 100 and 18. We use similar-based prompt retrieval for all methods and report the average results with 3 different runs for each method. We can observe that our method works better than Random and Vote-$k$ in almost all cases (17/18) under two annotation budgets. The best result in each case is **bolded**. We also provide the maximum and minimum values of the results in Appendix C.3.

**Measurement on performance.** We first perform the evaluations for Random, Vote-$k$, and our method. The annotation budget is set to 18 and 100 respectively following the same setting as Vote-$k$. Note that we include 18 as the annotation budget considering all annotated examples can be fit to the prompt of the large language models within context limits. Therefore, the prompt retrieve stage can be ignored and the evaluation results can naturally represent the quality of the selected examples. We provide experimental results in Table 1. As can be seen, our method achieves better performance than baselines

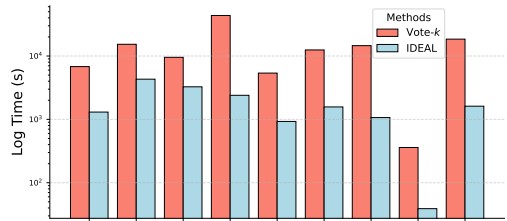

Figure 3: Comparison of our method and Vote-$k$ with respect to time consumption during subset selection under the same hardware condition. Here the annotation budget is 18. The y-axis represents the time consumption with a *log scale*. We can observe that our method largely reduces the time cost compared with Vote-$k$.

---

[3]https://github.com/HKUNLP/icl-selective-annotation.

in most of the evaluation scenarios (17 out of 18). Interestingly, we find that random selection outperforms Vote-$k$ in 3 out of 18 cases. We conjecture that, under some ideal circumstances, the selected subset by random selection can approximate the distribution of full data. If test data follows the same distribution, good performance can be achieved. Note that we also illustrate selected examples and label distributions in selective annotations in Appendix C.1 and Appendix C.4 respectively.

**Measurement on time cost.** Previous work Vote-$k$ (Su et al., 2023) encompasses generating prediction for most unlabeled data with a set of selected examples as prompts and performs data selection according to the confidence scores of the prediction. However, this process results in large unnecessary costs at inference time. Meanwhile, LLMs are often used as a service and an extra charge will appear with the usage of the token in both the input and output. In Figure 3, we compare the time cost of subset selection in our method against Vote-$k$ on all tasks with the same hardware. The annotation budget is set to 18. We can observe that our method saves a tremendous amount of cost compared to Vote-$k$. Specifically, under the same hardware conditions, IDEAL achieves a $7.8\times$ lead on average over Vote-$k$. The speed improvement benefits from the fact that the proposed method does not need to perform example selection by generating predictions on a large number of unlabeled examples and is completely unsupervised.

## 4.3 MORE ANALYSIS

### 4.3.1 LARGER INFLUENCE BRINGS BETTER PERFORMANCE

We conduct experiments to investigate the correlation between subset influence and its corresponding in-context learning performance. Specifically, we randomly select a collection of example subsets from a large unlabeled data pool. We then evaluate each subset as a prompt and record its performance and influence in the constructed graph, resulting in a set of influence-performance pairs. Our goal is to analyze the correlation between these two metrics. To achieve this, we perform experiments on SST-5 and MNLI. We sample 30 subsets and order them according to their influences, where each subset includes 5 examples. We divide this sorted subset sequence equally into three influence levels, with each level containing 10 subsets. We visualize the performance of subsets in each influence level in Figure 4. Our analysis reveals that subsets with larger influence levels achieve better average, median, and worst-case performance. This finding further demonstrates that quantifying the influence of each potential subset is an effective metric in the selective annotation problem.

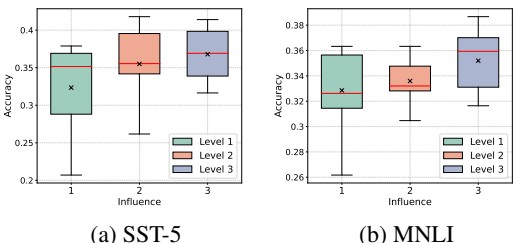

(a) SST-5       (b) MNLI

Figure 4: Influence vs. Performance. The illustration of the positive correlation between the influence achieved by Algorithm 1 and final performance.

### 4.3.2 COMPARISONS WITH ALTERNATIVE METHODS

We also compare our method with other alternative methods that can select the coreset from large-scale unlabeled data. We perform the evaluations on MRPC, MNLI and HellaSwag. We include the following alternative methods (1) $K$-Means (Lloyd, 1982), which groups all examples into $m$ clusters, and selects the centroid example from each cluster. (2) Maximizing facility location (MFL) (Lin & Bilmes, 2009), which aims at optimizing the representativeness of the selected subset. (3) Fast Vote-$k$ (Su et al., 2023), which is an efficient alternative to Vote-$k$ which directly picks $m$ examples with the largest Vote-$k$ scores.

| Method | $K$-Means | MFL | Fast Vote-$k$ | Vote-$k$ | IDEAL |
|---|---|---|---|---|---|
| MRPC | 57.4 | 58.2 | 59.3 | 61.1 | **63.0** |
| MNLI | 35.8 | 38.8 | 39.5 | 39.1 | **40.0** |
| HellaSwag | 65.4 | 65.2 | 65.9 | 66.5 | **67.1** |

Table 2: Comparisons of alternative methods that can select a coreset from large-scale unlabeled data. The annotation budget is 18. Experimental results are reported by averaging over three random trials. The performance of the baseline Vote-$k$ is also included here. The best performance in each case is **bolded**.

| Method | | Datasets | | |
|---|---|---|---|---|
| Selection | Retrieval | MRPC | MNLI | HellaSwag |
| Vote-$k$ | Similar | 64.6 | 38.9 | 67.9 |
| IDEAL | Similar | **66.4** | **41.0** | **68.6** |
| Vote-$k$ | Random | 60.7 | 37.8 | 64.6 |
| IDEAL | Random | **62.5** | **39.0** | **66.8** |

| Method | Models | Test Domain | |
|---|---|---|---|
| | | IMDb | BoolQ Cst. |
| Vote-$k$ | GPT-Neo | 71.1 | 56.4 |
| IDEAL | GPT-Neo | **72.2** | **58.0** |
| Vote-$k$ | GPT-J | 76.4 | 56.1 |
| IDEAL | GPT-J | **76.8** | **56.4** |

Table 3: Comparison of random and similar prompt retrieval with Vote-$k$ and IDEAL on MRPC, MNLI, and HellaSwag. The subset selection method with a similar prompt retrieve achieves better performance compared with its version with a random prompt retrieve method. The best performance in each case is **bolded**.

Table 4: The evaluations on out-of-distribution tasks. We show the performance of different methods on IMDb and BoolQ Contrast Set (target domains). In the evaluations, the prompts consist of selected SST-2 and BoolQ training examples, respectively (source domains). The best performance in each case is **bolded**.

We show the results in Table 2. We can observe IDEAL consistently outperforms the baselines in all datasets, demonstrating its superiority. Note that, the graph-based methods (Vote-$k$, Fast Vote-$k$, and our IDEAL) outperform the methods non-graph-based methods ($K$-Means and MFL) in all cases. This phenomenon suggests that graph-based methods are suitable for capturing similarity relationships between examples in the selective annotation problem, which can lead to better results.

### 4.3.3 EVALUATION WITH DIFFERENT RETRIEVAL METHODS

In previous experiments, we used a similarity-based prompt retrieval method by default. In this section, we conduct experiments to quantify the effect of different prompt retrieval methods under the annotation 100. We present the results in Table 3. We observe that both Vote-$k$ and IDEAL suffer from a significant performance drop when the prompt retrieval method is changed from similarity-based to random selection. Notably, IDEAL also achieves better performance than Vote-$k$ when combined with random retrieval in all datasets. It suggests that IDEAL can cultivate a more stable training subset (Chang & Jia, 2023) for in-context learning tasks. Note that we also show that our IDEAL is more stable and robust against the order of in-context examples in Appendix C.5.

### 4.3.4 EVALUATION ON OTHER LANGUAGE MODELS

Here we evaluate IDEAL on other language models, including GPT-Neo 2.7B (Black et al., 2021), and the advanced chat model GPT-3.5-Turbo where we use the same instruction as other language models for each dataset. While GPT-3.5-Turbo has mainly been optimized for chat, it also performs well on traditional completion tasks (Kheiri & Karimi, 2023). To conduct experiments, we select three classification tasks (MRPC, MNLI, and RTE), considering they are easier for prompting GPT-3.5-Turbo to return responses without pleasantries or explanatory content.

The evaluation results are presented in Figure 5. Our evaluations reveal that IDEAL consistently outperforms the baselines across all models tested. This demonstrates the versatility of our method in the context of learning tasks using models of varying sizes. Notably, we observe that the largest model, i.e., GPT-3.5-Turbo, performs worse than GPT-Neo and GPT-J. This situation arises because GPT-3.5-Turbo is primarily optimized for chat tasks and faces challenges in following human instructions for classification. This scenario also has been identified in Ye et al. (2023).

### 4.3.5 EVALUATION ON OUT-OF-DISTRIBUTION TASKS

We further evaluate our method on out-of-distribution tasks (Zhou et al., 2022; Wang et al., 2022b; Zhang et al., 2023b; Huang et al., 2023c;d), where there is a distribution shift between the selective annotation data and test data. Following (Chang & Jia, 2023), we compare IDEAL and Vote-$k$ using SST-2 (Socher et al., 2013), BoolQ (Clark et al., 2019) as source tasks, and IMDb (Maas et al., 2011), BoolQ Contrast Set (Gardner et al., 2020) as target tasks, respectively. In all evaluations, we set the annotation budget as 18 and use the similarity-based retrieve to perform the evaluations on the test set in target domains. We use GPT-J 6B and GPT-Neo 2.7B here and show the results in

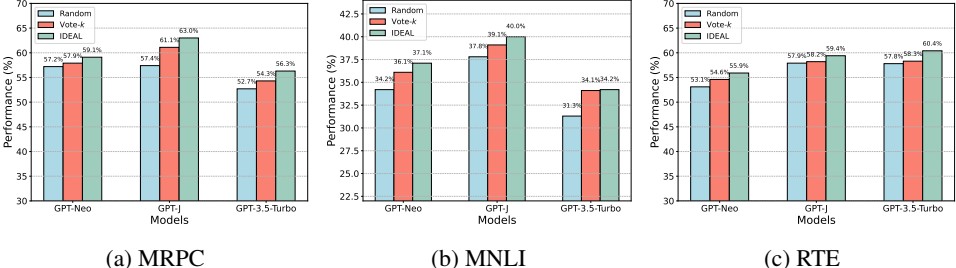

| (a) MRPC | (b) MNLI | (c) RTE |

Figure 5: Comparisons with various models when the annotation budget is 18. IDEAL consistently achieves the best performance compared with baselines in models with different datasets.

Table 4. We can observe that IDEAL still outperforms baselines on all datasets with two models, implying that IDEAL could select the subset which could depict the invariant properties of this kind of tasks and generalize to out-of-distribution scenarios.

### 4.4 CASE STUDY: AUTOMATIC ANNOTATION

In previous experiments, we used a small set of manually annotated examples as candidate prompts to make predictions. In contrast, here we are interested in a case study that utilizes the subset selected by IDEAL to annotate all available unlabeled data automatically, leading to a larger set of candidate prompts. Specifically, we first choose an initial subset from the pool of unlabeled data using IDEAL and manually label this selected subset. Afterward, we simulate the information diffusion process from the initial subset to all other data, where we employ those activated data as prompts to predict upcoming activated data at each step and label them accordingly with prediction results. This process ultimately

| Method | MRPC | SST-5 | MNLI | DBpedia | RTE |
|---|---|---|---|---|---|
| Vote-$k$ | 63.8 | 48.6 | 39.5 | 90.2 | 55.7 |
| IDEAL | 65.2 | 49.4 | **40.3** | 90.8 | 57.4 |
| Auto-IDEAL | **65.8** | **50.4** | 39.8 | **91.8** | **58.3** |

Table 5: Comparison between Vote-$k$, IDEAL, and Auto-IDEAL. Auto-IDEAL is an expanded version of IDEAL for automatic annotation. We evaluate these algorithms on all classification tasks and average their performance over three random trials. The best performance in each case is **bolded**. The results indicate that Auto-IDEAL can enhance the performance of IDEAL and achieve the best performance in 4 out of 5 cases.

makes a fully labeled training dataset. Finally, all examples (including manual labeling and automatic labeling ) are utilized as potential prompts in conjunction with the prompt retrieve technique for final testing. We name this paradigm as Auto-IDEAL and compare it with Vote-$k$ and origin IDEAL on all classification datasets. We choose 300 training data for each dataset to perform experiments. The manual annotation budget is set to 150, i.e., half of the labels of the candidate prompts in Auto-IDEAL are annotated automatically. Experimental results are provided in Table 5. As can be observed, Auto-IDEAL even achieves better performance than IDEAL in 4 of 5 cases. Notably, although the performance is worse on MNLI, it is still competitive (better than Vote-$k$). It suggests that expanding the candidate prompts through automatic annotation following the diffusion process can further boost the performance of IDEAL. It benefits from the fact that information only diffuses between similar examples. Therefore, unlabeled examples will be automatically annotated using the most similar annotated examples as prompts leading to a promising annotation success rate.

## 5 CONCLUSION

A series of recent works have confirmed the powerful ability of in-context learning for large language models. We investigate the ability from the perspective of selective annotations and propose an influence-driven method that selects a subset of data that acts as a proxy and closely approximates full data. Theoretical analysis is provided to establish an upper limit for the global optimal solution, and demonstrate that our greedy search algorithm selects a subset with influence at least as substantial as a specific proportion of the optimal solution's influence. Empirical evaluations illustrate the superiority of our method across a range of benchmarks, delivering superior performance while largely reducing the time required for subset selection. We hope this work can help researchers and practitioners understand the promise and potential of selective annotations in in-context learning, and facilitate them in the efficient conceptualization of novel language-based challenges.

## ACKNOWLEDGEMENTS

Tongliang Liu is partially supported by the following Australian Research Council projects: FT220100318, DP220102121, LP220100527, LP220200949, and IC190100031.

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

# A    RELATED WORK

## A.1    IN-CONTEXT LEARNING

In-context learning (ICL) has become a new paradigm for natural language processing (NLP), where large language models make predictions only based on contexts augmented with a few examples (Dong et al., 2023; Xie et al., 2022; Shin et al., 2022; Zhang et al., 2023a; Bai et al., 2023; Huang et al., 2023b). A series of works attempts to revise, enhance, and understand ICL, which include but are not limited to prompt tuning (Kim et al., 2022; Wang et al., 2022a; Mishra et al., 2022), analyzing intrinsic mechanism (Bansal et al., 2022; Chan et al., 2022; Li et al., 2023a; Garg et al., 2022), evaluations (Srivastava et al., 2023; Wang et al., 2022c), applications in multiple domains (Chen et al., 2022; Lee et al., 2022; Cho et al., 2023), and etc.

Despite in-context learning has shown impressive performance in various domains, its efficacy is sensitive to the selection of in-context learning examples (Zhao et al., 2021b; Lu et al., 2021). Considering this, multiple methods have been proposed to select the optimal in-context learning examples to achieve optimal performance. These methods retrieve the most relevant examples subset with/without specific order for each query (Wu et al., 2022; Liu et al., 2021; Gao et al., 2020) Alternatively, some methods aim to find a set of examples once for all queries on the same task (Li & Qiu, 2023; Diao et al., 2023). Specifically, (Wu et al., 2022) formally defines the problem of self-adaptive In-context learning, which aims to search for the best In-context learning examples and corresponding order for each input query. While, Li & Qiu (2023); Diao et al. (2023) focus on finding one fixed set of examples for each task. However, these methods rely on the assumption that large-scale annotated training examples are always available.

Different from them, this paper studies selective annotations for ICL, which can effectively reduce the annotation cost in ICL. Furthermore, compared with recent work (Su et al., 2023), as discussed in the main paper, this work is superior in many aspects, such as the end-to-end manner, mitigation of the trade-off between diversity and representativeness, theoretical guarantees, and better empirical performance.

### A.2 Coreset selection

Coreset selection focuses on selecting a small but highly informative subset from a large dataset for follow-up tasks, which can significantly reduce the data storage cost and training consumption (Huang et al., 2018; 2023a; Feldman & Zhang, 2020; Sorscher et al., 2022; Xia et al., 2023b; Li et al., 2023b; Xia et al., 2022). Most of the works on coreset selection target the scenes of supervised learning and classification (Sener & Savarese, 2018; Toneva et al., 2019; He et al., 2023). Only a few works extend coreset selection into unsupervised cases (Sorscher et al., 2022; Su et al., 2023). This paper studies unsupervised data selection for annotations in ICL, which reduces the annotation expenses of prompts and helps large language models become better few-shot learners. Also, it enjoys theoretical support. Therefore, this work is different from previous efforts and contributes to the research community.

### A.3 Data distillation

Data distillation (Wang et al., 2018; Zhao et al., 2021a; Shin et al., 2023; Cui et al., 2023; Du et al., 2023; Loo et al., 2023) is an alternative approach for dataset compression and curation, which is inspired by knowledge distillation which could be categorized into efficient ML (Ma et al., 2023a;b; Zheng et al., 2021; Zhang et al., 2021; Zheng et al., 2023). Different from coreset selection, this series of works target *synthesizing* a small but informative dataset as an alternative to the original dataset. However, data distillation is criticized for only synthesizing a small number of data points due to computational source limitations (Xia et al., 2023a; Yang et al., 2023). The performances of data distillation and data selection are therefore not compared directly. Besides, it is under-explored about how to perform data distillation in an unsupervised manner on natural language processing tasks. Based on this analysis, the data distillation strategy is not involved in empirical evaluations.

## B Proofs

### B.1 Preliminary theoretical results

We first present some preliminary theoretical results and their corresponding proofs for the subsequential proofs of Proposition 1 and Theorem 1.

#### B.1.1 Lemma 1

**Lemma 1.** *Given a graph $\mathcal{G} = (\mathbf{V}, \mathbf{E}, \mathbf{P})$, if the influence function meets Condition 1, then for $\forall \mathcal{S}_i, \mathcal{S}_j \subseteq \mathbf{V}$:*

$$f_{\mathcal{G}}(\mathcal{S}_i) - f_{\mathcal{G}}(\mathcal{S}_j) \leq \sum_{\mathbf{v} \in \mathcal{S}_i - \mathcal{S}_j} \psi_{\mathbf{v}}(\mathcal{S}_j) - \sum_{\mathbf{v} \in \mathcal{S}_j - \mathcal{S}_i} \psi_{\mathbf{v}}(\mathcal{S}_i \cup \mathcal{S}_j - \mathbf{v}), \tag{5}$$

*where $\psi_{\mathbf{v}}(\mathcal{S}_j) := f_{\mathcal{G}}(\mathcal{S}_j \cup \mathbf{v}) - f_{\mathcal{G}}(\mathcal{S}_j)$.*

*Proof.* The proof is inspired by (Rolnick & Weed, 2014). We first let

$$\mathcal{S}_i - \mathcal{S}_j = \{\mathbf{a}_1, ..., \mathbf{a}_r\} \tag{6}$$

and

$$\mathcal{S}_j - \mathcal{S}_i = \{\mathbf{b}_1, ..., \mathbf{b}_q\}, \tag{7}$$

where $r \in \mathbb{N}_+$ and $q \in \mathbb{N}_+$. According to Eq. (6), for the subsets $\mathcal{S}_i$ and $\mathcal{S}_j$, we have

$$\mathcal{S}_j \cup \mathcal{S}_i = \mathcal{S}_j \cup \{\mathbf{a}_1, ..., \mathbf{a}_r\}. \tag{8}$$

Afterward, we obtain

$$f_{\mathcal{G}}(\mathcal{S}_j \cup \mathcal{S}_i) - f_{\mathcal{G}}(\mathcal{S}_j) = f_{\mathcal{G}}(\mathcal{S}_j \cup \{\mathbf{a}_1, ..., \mathbf{a}_r\}) - f_{\mathcal{G}}(\mathcal{S}_j). \tag{9}$$

At a high level, Eq. (9) is to calculate the influence improvement of $\mathcal{S}_j$ after adding data points $\{\mathbf{a}_1, ..., \mathbf{a}_r\}$ into $\mathcal{S}_j$. As the influence improvement of adding one sequence of data points is equal to the sum of the influence improvement at each step, we have,

$$f_{\mathcal{G}}(\mathcal{S}_j \cup \mathcal{S}_i) - f_{\mathcal{G}}(\mathcal{S}_j) \tag{10}$$

$$= f_{\mathcal{G}}(\mathcal{S}_j \cup \mathbf{a}_1) - f_{\mathcal{G}}(\mathcal{S}_j) + \sum_{k=2}^{r} [f_{\mathcal{G}}(\mathcal{S}_j \cup \{\mathbf{a}_1, ..., \mathbf{a}_k\}) - f_{\mathcal{G}}(\mathcal{S}_j \cup \{\mathbf{a}_1, ..., \mathbf{a}_{k-1}\})]$$

$$= \psi_{\mathbf{a}_1}(\mathcal{S}_j) + \sum_{k=2}^{r} \psi_{\mathbf{a}_k}(\mathcal{S}_j \cup \{\mathbf{a}_1, ..., \mathbf{a}_{k-1}\}).$$

Under Condition 1, as $\mathcal{S}_j \subset \mathcal{S}_j \cup \{\mathbf{a}_1, ..., \mathbf{a}_{k-1}\}$, we have

$$f_{\mathcal{G}}(\mathcal{S}_j \cup \mathcal{S}_i) - f_{\mathcal{G}}(\mathcal{S}_j) = \psi_{\mathbf{a}_1}(\mathcal{S}_j) + \sum_{k=2}^{r} \psi_{\mathbf{a}_k}(\mathcal{S}_j \cup \{\mathbf{a}_1, ..., \mathbf{a}_{k-1}\}) \tag{11}$$

$$\leq \sum_{k=1}^{r} \psi_{\mathbf{a}_k}(\mathcal{S}_j) = \sum_{\mathbf{a} \in \mathcal{S}_i - \mathcal{S}_j} \psi_{\mathbf{a}}(\mathcal{S}_j).$$

Similarly,

$$f_{\mathcal{G}}(\mathcal{S}_j \cup \mathcal{S}_i) - f_{\mathcal{G}}(\mathcal{S}_i) \tag{12}$$

$$= \psi_{\mathbf{b}_1}(\mathcal{S}_i) + \sum_{k=2}^{q} \psi_{\mathbf{b}_k}(\mathcal{S}_i \cup \{\mathbf{b}_1, ..., \mathbf{b}_{k-1}\}) \geq \sum_{k=1}^{q} \psi_{\mathbf{b}_k}(\mathcal{S}_i \cup \mathcal{S}_j - \mathbf{b}_k) = \sum_{\mathbf{b} \in \mathcal{S}_j - \mathcal{S}_i} \psi_{\mathbf{b}}(\mathcal{S}_i).$$

By subtracting (12) from (10), we have

$$f_{\mathcal{G}}(\mathcal{S}_i) - f_{\mathcal{G}}(\mathcal{S}_j) \leq \sum_{\mathbf{v} \in \mathcal{S}_i - \mathcal{S}_j} \psi_{\mathbf{v}}(\mathcal{S}_j) - \sum_{\mathbf{v} \in \mathcal{S}_j - \mathcal{S}_i} \psi_{\mathbf{v}}(\mathcal{S}_i \cup \mathcal{S}_j - \mathbf{v}). \tag{13}$$

$\square$

### B.1.2 LEMMA 2

**Lemma 2.** *Given a graph $\mathcal{G} = (\mathbf{V}, \mathbf{E}, \mathbf{P})$, for any subset $\mathcal{S} \subset \mathbf{V}$ and any $\mathbf{v} \in \mathbf{V}$, the influence function $f_{\mathcal{G}}$ satisfies*

$$\psi_{\mathbf{v}}(\mathcal{S}) = f_{\mathcal{G}}(\mathcal{S} \cup \mathbf{v}) - f_{\mathcal{G}}(\mathcal{S}) \geq 0 \tag{14}$$

*Proof.* We consider two cases to finish the proof.

*Case 1* ($\mathbf{v} \in \mathbf{V} \wedge \mathbf{v} \notin \mathcal{S}$). In this case, the influence improvement is at least 1 since $\mathbf{v}$ itself has been included, i.e.,

$$\psi_{\mathbf{v}}(\mathcal{S}) = f_{\mathcal{G}}(\mathcal{S} \cup \mathbf{v}) - f_{\mathcal{G}}(\mathcal{S}) \geq 1. \tag{15}$$

*Case 2* ($\mathbf{v} \in \mathbf{V} \wedge \mathbf{v} \in \mathcal{S}$). In this case, the influence improvement is 0 since $\mathbf{v}$ has already been included in $\mathcal{S}$, i.e.,

$$\psi_{\mathbf{v}}(\mathcal{S}) = f_{\mathcal{G}}(\mathcal{S} \cup \mathbf{v}) - f_{\mathcal{G}}(\mathcal{S}) = 0. \tag{16}$$

Combining the above two cases, we conclude that, for $\forall \mathbf{v} \in \mathbf{V}$, the influence function $f_{\mathcal{G}}$ satisfies

$$f_{\mathcal{G}}(\mathcal{S} \cup \mathbf{v}) - f_{\mathcal{G}}(\mathcal{S}) \geq 0. \tag{17}$$

$\square$

### B.2 PROOF OF PROPOSITION 1

*Proof.* Given a graph $\mathcal{G} = (\mathbf{V}, \mathbf{E}, \mathbf{P})$, for $\forall \mathcal{S}_i, \mathcal{S}_j \subset \mathbf{V}$, according to Lemma 2, we have

$$\sum_{\mathbf{v} \in \mathcal{S}_i - \mathcal{S}_j} \psi_{\mathbf{v}}(\mathcal{S}_i \cup \mathcal{S}_j - \mathbf{v}) \geq 0. \tag{18}$$

Taking (18) into Lemma 1, we have

$$f_{\mathcal{G}}(\mathcal{S}_i) - f_{\mathcal{G}}(\mathcal{S}_j) \leq \sum_{\mathbf{v} \in \mathcal{S}_i - \mathcal{S}_j} \psi_{\mathbf{v}}(\mathcal{S}_j). \tag{19}$$

We use $\mathcal{S}_m^*$ to denote the optimal solution as discussed in the main paper. At any step $t$ in Algorithm 2, we substitute $\mathcal{S}_m^*$ (resp. $\mathcal{S}_t$) into $\mathcal{S}_i$ (resp. $\mathcal{S}_j$) in (19), we can derive

$$f_{\mathcal{G}}(\mathcal{S}_m^*) \leq f_{\mathcal{G}}(\mathcal{S}_t) + \sum_{\mathbf{v} \in \mathcal{S}_m^* - \mathcal{S}_t} \psi_{\mathbf{v}}(\mathcal{S}_t). \tag{20}$$

According to Condition 1,

$$\psi_{\mathbf{v}}(\mathcal{S}_t) \geq \psi_{\mathbf{v}}(\mathcal{S}_{t+1}) \tag{21}$$

holds. Taking both (20) and (21) into (19), we have for any $t$,

$$f_{\mathcal{G}}(\mathcal{S}_m^*) \leq f_{\mathcal{G}}(\mathcal{S}_t) + m\psi_{t+1}. \tag{22}$$

$\square$

### B.3 PROOF OF THEOREM 1

*Proof.* Recall that

$$\psi_t = f_{\mathcal{G}}(\mathcal{S}_t) - f_{\mathcal{G}}(\mathcal{S}_{t-1}). \tag{23}$$

According to Proposition 1, we have

$$f_{\mathcal{G}}(\mathcal{S}_m^*) - f_{\mathcal{G}}(\mathcal{S}_t) \leq m\psi_{t+1} = m(f_{\mathcal{G}}(\mathcal{S}_{t+1}) - f_{\mathcal{G}}(\mathcal{S}_t)). \tag{24}$$

Afterwards, (24) equals to,

$$f_{\mathcal{G}}(\mathcal{S}_m^*) - f_{\mathcal{G}}(\mathcal{S}_t) - (f_{\mathcal{G}}(\mathcal{S}_m^*) - f_{\mathcal{G}}(\mathcal{S}_{t+1})) \geq \frac{1}{m}(f_{\mathcal{G}}(\mathcal{S}_m^*) - f_{\mathcal{G}}(\mathcal{S}_t)) \tag{25}$$

$$\iff f_{\mathcal{G}}(\mathcal{S}_m^*) - f_{\mathcal{G}}(\mathcal{S}_{t+1}) \leq \frac{m-1}{m}(f_{\mathcal{G}}(\mathcal{S}_m^*) - f_{\mathcal{G}}(\mathcal{S}_t)).$$

Based on (25), we have

$$f_{\mathcal{G}}(\mathcal{S}_m^*) - f_{\mathcal{G}}(\mathcal{S}_{t+1}) \leq \frac{m-1}{m}(f_{\mathcal{G}}(\mathcal{S}_m^*) - f_{\mathcal{G}}(\mathcal{S}_t)) \tag{26}$$

$$\leq (\frac{m-1}{m})^2(f_{\mathcal{G}}(\mathcal{S}_m^*) - f_{\mathcal{G}}(\mathcal{S}_{t-1}))$$

$$\leq ... \leq (\frac{m-1}{m})^{t+1}(f_{\mathcal{G}}(\mathcal{S}_*^m) - f_{\mathcal{G}}(\mathcal{S}_0)).$$

Since $f_{\mathcal{G}}(\mathcal{S}_0) = f_{\mathcal{G}}(\emptyset) = 0$, we have

$$\frac{f_{\mathcal{G}}(\mathcal{S}_m^*) - f_{\mathcal{G}}(\mathcal{S}_{t+1})}{f_{\mathcal{G}}(\mathcal{S}_m^*)} \leq (\frac{m-1}{m})^{t+1}. \tag{27}$$

When Algorithm 2 terminates at step $t = m - 1$, we have,

$$f_{\mathcal{G}}(\mathcal{S}_m) \geq (1 - (1 - 1/m)^m)f_{\mathcal{G}}(\mathcal{S}_m^*). \tag{28}$$

$\square$

# C  Supplementary Experimental Results

## C.1  Selected examples

In Table 6, for illustration purposes, we provide a few examples from the selection by our method, when the annotation size is 18.

| Dataset | Input |
|---|---|
| MRPC | a. Input: The two Democrats on the five-member FCC held a press conference to sway opinion against [...] 
   Output: not equivalent 
 a. Input: The report shows that drugs sold in Canadian pharmacies are manufactured in facilities approved by Health Canada [...] 
   Output: equivalent 
 c. Input: The chief merchandising officer decides what the store is going to sell [...] 
   Output: equivalent |
| SST-5 | a. Input: plodding, poorly written, murky and weakly acted, the picture feels as if everyone making it lost their movie mojo. 
   Output: very negative 
 b. Input: duvall is strong as always . 
   Output: very positive 
 c. Input: lohman adapts to the changes required of her , but the actress and director peter kosminsky never get the audience to break [...] 
   Output: neutral |
| MNLI | a. Input: This prosperous city has many museums, including a well-endowed Musee des Beaux-Arts (Square Verdrel) [...] 
   Output: False 
 b. Input: Duhame, who today makes her living as a graphic designer and illustrator, calls her book [...] 
   Output: Inconclusive 
 c. Input: At the agency or program level, it included management's public commitment to reduce fraud and errors, as. Based on that information [...] 
   Output: True |
| DBpedia | a. Input: Lars Nielsen (born 3 November 1960 in Copenhagen) is a Danish rower. 
   Output: athlete 
 b. Input: Calhoun County High School is a public secondary school in St. Matthews South Carolina USA. 
   Output: educational institution 
 c. Input: David Goldschmid (sometimes credited as Dave Goldschmid) is an American television writer and producer currently writing for the daytime drama General Hospital. 
   Output: artist |
| RTE | a. Input: In sub-Saharan Africa about one in every 30 people is infected with HIV.. 30% of the people infected with HIV live in Africa.. 
   Output: False 
 b. Input: The drawbacks of legalization do not imply that our current version of prohibition is the optimal drug strategy; it may well [...] 
   Output: False 
 c. Input: For example, the fields of Western farmers feed the United States and many other parts of the world, and India's irrigation [...] 
   Output: True |
| HellaSwag | a. Input: The topic is Preparing salad. An illustrated egg, the website "startcooking com" and "vegetable salad" [...] 
   Output: is shown from above. 
 b. Input: The topic is Pets and Animals. [header] How to treat an injured rabbit's paw [title] Identify sore hocks. [step] Pododermatitis [...] 
   Output: Once the condition has set in, though, you'll need to take quick action to treat the injury. Leaving [...] 
 c. Input: The topic is Playing squash. Two men stand on a racquetball court. the men 
   Output: stretch then begin playing. |

Table 6: For illustration purposes, under our method, we show randomly selected three examples from each of the six datasets in one same run (excluding the other three datasets due to their length) when the annotation budget is set to 18.

## C.2 VISUALIZATION OF SELECTED EXAMPLES

Here we provide a umap (McInnes et al., 2018) visualization of selected examples. To avoid the denseness, we choose the annotation budget as 5. The visualization can be checked in Figure 6. First, comparing subfigures (a) and (b), we can clearly see that the selection of Vote-$k$ is much biased, and our IDEAL can identify a subset that is more favorable to be a proxy of full data. Second, comparing subfigures (c) and (d), we can see that the selected subset by Vote-$k$ is distributed on the right of full data. By comparison, our IDEAL can select a subset that is distributed more uniformly.

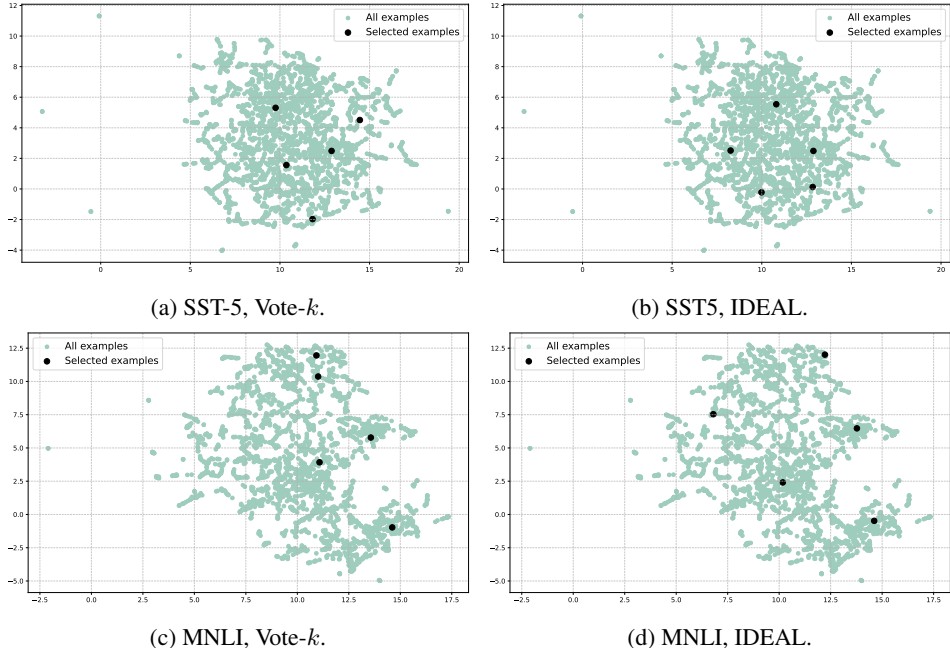

(a) SST-5, Vote-$k$.  (b) SST5, IDEAL.

(c) MNLI, Vote-$k$.  (d) MNLI, IDEAL.

Figure 6: Umap (McInnes et al., 2018) visualization to compare five selected examples from all examples using fully unsupervised methods Vote-$k$ and IDEAL (ours). Compared with Vote-$k$, IDEAL could choose the examples to better represent the whole data rather than get involved in diversity and including outliers.

## C.3 DETAILED EXPERIMENTAL RESULTS IN TABLE 1

|     | Method | MRPC | SST-5 | MNLI | DBpedia | RTE |
|-----|--------|------|-------|------|---------|-----|
| 100 | Random | 64.3/68.4/58.6 | 49.6/51.1/47.2 | 38.2/40.2/36.7 | 89.8/91.0/88.2 | 55.3/55.9/55.1 |
| 100 | Vote-$k$ | 64.6/68.8/62.1 | 46.6/47.2/46.1 | 38.9/43.8/35.5 | 89.2/89.8/88.7 | 57.6/58.2/57.4 |
| 100 | IDEAL | **66.4**/67.9/64.8 | **51.4**/53.5/49.6 | **41.0**/41.4/40.2 | **90.6**/91.4/89.5 | **58.9**/60.9/57.4 |
| 18  | Random | 57.4/68.8/39.8 | 42.9/46.9/39.1 | 37.8/39.4/35.2 | 85.2/87.5/83.9 | 57.9/58.9/57.0 |
| 18  | Vote-$k$ | 61.1/67.2/52.7 | 41.7/45.7/37.1 | 39.1/43.8/32.0 | 89.9/94.1/87.1 | 58.2/58.9/57.8 |
| 18  | IDEAL | **63.0**/63.7/62.5 | **43.2**/45.7/39.5 | **40.0**/41.8/37.1 | **90.1**/90.2/89.8 | **59.4**/60.9/57.8 |

Table 7: Mean/maximum/minimum evaluation results of all methods on classification tasks in Table 1 over three different trials. The best mean result in each case is **bolded**.

In the main paper (Table 1), we report the mean evaluation results for different methods over three random trials. Here we provided the detailed results of Table 1 with mean/maximum/minimum values. We can observe IDEAL achieves stable results compared with baselines. Moreover, the worst-case performance of IDEAL is obviously better compared with baselines in most cases.

| | Method | HellaSwag | MWoZ | GeoQ | Xsum |
|---|---|---|---|---|---|
| 100 | Random | 66.7/70.3/64.1 | 39.9/48.4/39.9 | 55.3/57.8/53.1 | 15.3/16.4/14.8 |
| 100 | Vote-$k$ | 67.9/69.9/64.0 | 48.3/50.8/46.9 | **58.8**/60.5/57.0 | 17.2/17.6/16.4 |
| 100 | IDEAL | **68.6**/71.9/65.2 | **52.2**/55.9/49.1 | 58.2/60.5/54.7 | **19.9**/20.2/19.5 |
| 18 | Random | 66.0/68.8/63.7 | 37.0/46.5/28.1 | 47.5/49.2/44.9 | 13.6/14.5/12.5 |
| 18 | Vote-$k$ | 66.5/71.9/62.5 | 37.7/43.8/32.4 | 50.9/54.3/47.7 | 15.2/16.0/14.5 |
| 18 | IDEAL | **67.1**/71.9/64.5 | **38.5**/47.3/30.9 | **52.0**/53.9/50.8 | **19.6**/20.2/18.9 |

Table 8: Mean/maximum/minimum evaluation results of all methods on multi-choice, dialogue, and generation tasks in Table 1 over three different trials. The best mean result in each case is **bolded**.

| Method | MRPC | | MNLI | | | RTE | |
|---|---|---|---|---|---|---|---|
| | Equivalent | Not equivalent | True | Inconclusive | False | True | False |
| Original | 2023 | 977 | 1051 | 965 | 984 | 1241 | 1249 |
| Random | 70 | 30 | 30 | 39 | 31 | 56 | 44 |
| Vote-$k$ | 64 | 36 | 27 | 35 | 38 | 46 | 54 |
| IDEAL | 65 | 35 | 37 | 34 | 29 | 49 | 51 |

Table 9: The numbers of different labels in the selected examples for different methods. "Original" denotes the label statistics of the original dataset. Under the annotation budget 100, IDEAL achieves the smallest ratio between the numbers of the most frequent class and the least frequent class in 2 out of 3 cases (MNLI and RTE), implying IDEAL can indeed mitigate the label skew problem.

| Method | MRPC | | SST-5 | | RTE | |
|---|---|---|---|---|---|---|
| | Mean | Std | Mean | Std | Mean | Std |
| Random | 44.2 | 0.02 | 45.4 | 0.02 | 57.3 | 0.02 |
| Vote-$k$ | 52.7 | 0.03 | 38.0 | 0.01 | 58.6 | 0.02 |
| IDEAL | 65.5 | 0.01 | 46.6 | 0.01 | 57.5 | 0.02 |

Table 10: The average performance of different methods by permuting the order of prompts for each test instance 10 times. We conduct experiments on MRPC, SST-5, and RTE datasets and report the average results with standard deviation. We can observe the subset selected by IDEAL achieves the best performance compared with baselines in 2 out of 3 cases. IDEAL also achieves the lowest standard deviations in all evaluations, which suggests IDEAL is a more stable and robust method against the order of prompts.

## C.4 LABEL DISTRIBUTIONS IN SELECTIVE ANNOTATIONS

Recall that the process of selective annotations is based entirely on similarities derived from sentence embeddings without labels. Therefore, we investigate whether the selected examples have label skew. Under an annotation budget of 100, we collect all selected examples in three classification tasks (MRPC, MNLI, and RTE) and show the numbers of different labels for different methods in Table 9. We also present the label statistics of the original training data. We observe that random selection shows a great variance. However, in an ideal case, it should achieve a similar distribution as the original training data. Notably, IDEAL achieves the smallest ratio between the numbers of the most frequent class and the least frequent class in 2 out of 3 cases (MNLI and RTE). This demonstrates that IDEAL can indeed balance the label distribution in the selected subset and mitigate the problem of label skew.

## C.5 PROMPT ORDER IN SELECTIVE ANNOTATION

As pointed out by (Lu et al., 2021), the performance of in-context learning is influenced not only by the selection of prompts but also by the order in which the prompts are presented to models. Although this work focuses solely on selective annotation problems, we are interested in explor-

ing whether the selected subset can still lead to better performance when the order of prompts is permuted. Under an annotation budget of 18, we first retrieve prompts for each test instance from selected subsets achieved by different selective annotation methods. We then permute the order of prompts for each test instance 10 times, resulting in 10 different experimental trials. We show the average performance of these 10 trials and make a comparison between different selective annotation methods. We conduct experiments on MRPC, SST-5, and RTE datasets and present the results in Table 10. The results show that IDEAL outperforms baselines in 2 out of 3 cases, suggesting that our method can choose more stable and robust subsets against changed prompt orders.

# D  SUPPLEMENTARY DESCRIPTIONS OF EXPERIMENTAL SETTINGS

## D.1  DETAILS OF DATASETS

In this paper, to demonstrate the superiority of our method, we employ 9 datasets typical datasets that have been widely used in previous NLP works (Su et al., 2023; Shi et al., 2023; Zhang et al., 2021) which can be categorized into 4 different tasks, including *classification* (MRPC (Dolan et al., 2004), SST-5 (Socher et al., 2013), MNLI (Williams et al., 2017), DBpedia (Lehmann et al., 2015), and RTE (Bentivogli et al., 2009)), *multi-choice* (HellaSwag (Zellers et al., 2019)), *dialogue* (MWoZ (Budzianowski et al., 2018)), and *generation* (GeoQuery (Zelle & Mooney, 1996) and Xsum (Narayan et al., 2018)). We list the datasets and the models used in Table 11.

|  | Datasets | Task | Models |
|---|---|---|---|
| **Classification** | MRPC (Dolan et al., 2004) | Paraphrase Detection | GPT-Neo, GPT-J, GPT-3.5-Turbo |
|  | SST-5 (Socher et al., 2013) | Sentiment Analysis | GPT-J |
|  | DBpedia (Lehmann et al., 2015) | Topic Classification | GPT-J |
|  | RTE (Bentivogli et al., 2009)) | Natural Language Inference | GPT-Neo, GPT-J, GPT-3.5-Turbo |
|  | MNLI (Williams et al., 2017) | Natural Language Inference | GPT-Neo, GPT-J, GPT-3.5-Turbo |
| **Multiple-Choice** | HellaSwag (Zellers et al., 2019) | Commonsense Reasoning | GPT-J |
| **Dialogue** | MWoZ (Budzianowski et al., 2018) | Dialogue State Tracking | Text-davinci-002 |
| **Generation** | GeoQuery (Zelle & Mooney, 1996) | Semantic Parsing | Text-davinci-002 |
|  | Xsum (Narayan et al., 2018) | Summarization | GPT-J |

Table 11: The datasets and corresponding models used in our experiments. We use GPT-J 6B and Text-davinci-002 by default. Other large language models are explored in §4.3.4.

To help readers better understand the datasets and tasks, for each of these datasets, we also list one example including both the input and output.

### D.1.1  MRPC

**Input**

```
Are the following two sentences 'equivalent' or 'not equivalent'?\nA
federal judge yesterday disconnected a new national \" do-not-call \"
 list , just days before it was to take effect , saying the agency
that created it lacked the authority ..\nA federal judge yesterday
struck down the national do-not-call registry slated to take effect
Oct. 1 , ruling the Federal Trade Commission had no authority to
create the list ..\nanswer:
```

**Output**

```
equivalent
```

### D.1.2  SST-5

**Input**

```
 How do you feel about the following sentence?\nsmug , artificial ,
ill-constructed and fatally overlong ... it never finds a consistent
tone and lacks bite , degenerating into a pious , preachy soap opera
.\nanswer:
```

**Output**

```
 neutral
```

### D.1.3   MNLI

**Input**

```
 yeah well the Cardinals i don't know  i think the Cowboys probably
have a a better team they just at the end of the season the kind of
got messed up with Aikman getting hurt because uh Laufenberg just
couldn't never really get it together at all of course he sat along
the sidelines all season he never really got in a game never did a
whole lot. Based on that information, is the claim The Cowboys should
 have started Laufenberg all season.  \"True\", \"False\", or \"
Inconclusive\"?\nanswer:
```

**Output**

```
 Inconclusive
```

### D.1.4   DBPEDIA

**Input**

```
 title: V\u00edctor David Loubriel; content:  V\u00edctor David
Loubriel Ort\u00edz is a Puerto Rican politician and former member of
 the Senate of Puerto Rico for the New Progressive Party (PNP).
Loubriel presented his candidacy for the Senate of Puerto Rico before
 2004. He ran for a candidate slot in the 2003 primaries obtaining
the most votes in his district (Arecibo).In the 2004 general election
 Loubriel won a seat in the 23rd Senate of Puerto Rico to represent
the district of Arecibo along with Jos\u00e9 Emilio Gonz\u00e1lez Vel
\u00e1zquez.
```

**Output**

```
 office holder
```

### D.1.5   RTE

**Input**

```
 MEXICO CITY (Reuters) - A deadly strain of swine flu never seen
before has broken out in Mexico, killing as many as 60 people and
raising fears it is spreading across North America. The World Health
Organization said it was concerned about what it called 800 \"
influenza-like\" cases in Mexico, and also about a confirmed outbreak
 of a new strain of swine flu in the United States. It said about 60
people had died in Mexico. Mexico's government said it had confirmed
that at least 16 people had died of the swine flu in central Mexico
and that there could be another 45 fatal victims..\nquestion: 800
Mexicans have been affected by a new form of swine influenza.. True
or False?\nanswer:
```

**Output**

```
 True
```

### D.1.6 HELLASWAG

**Input**

```
 The topic is Work World. [header] How to become a high school social
studies teacher [title] Obtain your bachelor's degree in education. [
step] All schools will require you to obtain at least your bachelor's
 degree in education. This degree will be proof that you are capable
of delivering information to students using the current educational
best practices.
```

**Output**

```
Make sure you've fully completed all of your course work and obtained
 your bachelor's degree before you seek certification or employment.
[substeps] Your electives should be based in social studies courses.
```

### D.1.7 MULTIWOZ

**Input**

```
CREATE TABLE hotel(
  name text,
  ......,
  internet text CHECK (internet IN (dontcare, yes, no))
)
/*
4 example rows:
SELECT * FROM hotel LIMIT 4;
name  pricerange  type  parking book_number_of_days book_day
book_people
area  stars internet
a and b guest house moderate guest house dontcare 3 friday 5 east 4
yes
......
/*
......
-- Using valid SQLite, answer the following multi-turn conversational
questions for the tables provided above.
Example #1
[context] hotel-area: west, hotel-stars: 3, hotel-internet: yes
[system] the hobsons house is available in that area .
Q: [user] that sounds like it will work . can i book that for 3
nights
starting wednesday ?
SQL: SELECT * FROM hotel WHERE book_day = wednesday AND book_people =
 1
AND book_number_of_days = 3 AND name = hobsons house;
......
```

**Output**

```
hotel WHERE book_day = wednesday AND book_number_of_days = 4 AND name
 =
warkworth house;
```

### D.1.8 GEOQ

**Input**

```
CREATE TABLE "border_info" ("state_name" text, "border" text)
/*
state_name    border
   alabama tennessee
   alabama   georgia
   alabama   florida
*/
```

```
......
-- Using valid SQLite, answer the following questions for the tables
provided above.
......
-- what is the longest river in the state with the highest point
SELECT
```

**Output**

```
RIVERalias0.RIVER_NAME FROM HIGHLOW AS HIGHLOWalias0, RIVER AS
RIVERalias0 WHERE HIGHLOWalias0.HIGHEST_ELEVATION = (SELECT MAX(
HIGHLOWalias1.HIGHEST_ELEVATION) FROM HIGHLOW AS HIGHLOWalias1 ) AND
RIVERalias0.TRAVERSE = HIGHLOWalias0.STATE_NAME ORDER BY RIVERalias0.
LENGTH DESC LIMIT 1;
```

### D.1.9  XSUM

**Input**

```
For decades, large numbers of Haitians have migrated – many of them
without papers – to the Dominican Republic, to escape the poverty and
 lack of employment in their homeland.\nIn 2013, the Dominican
Republic's highest court ruled that children born there to
undocumented migrants were not automatically eligible for Dominican
nationality.
......
 \nThere he strips the trees for firewood to make charcoal, to sell to
 Dominican traders for a few dollars.\nHe knows the practice damages
the fertility of the soil, but it's the only available source of
income.\n\"This is the only way we can survive,\" he says, motioning
at his family, stuck inside the world's forgotten migrant crisis.\
nYou can hear more of Will Grant's report on Heart and Soul on the
BBC World Service.
```

**Output**

```
 Immigration has long been a divisive issue on Hispaniola, the
Caribbean island shared by Haiti and the Dominican Republic.
```

### D.2  IMPLEMENTATION DETAILS

**General experimental conditions.** We primarily use PyTorch (Paszke et al., 2019) to implement our algorithm and baselines. For GPT-3.5-Turbo, we perform the experiments by calling the OpenAI API using a single Intel Xeon CPU. The GPT-J 6B and GPT-Neo 2.7B models are from the Huggingface transformer library (Wolf et al., 2019). We run all our experiments of GPT-J 6B and GPT-Neo 2.7B on a single NVIDIA Tesla V100 (32GB) GPU.

**Details of getting unlabeled data.** Since obtaining unlabeled examples in realistic scenarios is also a high-variance process, we follow the same setting as (Su et al., 2023) to simulate the realistic setting. We perform selective annotations from 3k instances that are randomly sub-sampled from training data for each task. For each experiment, we repeat the sub-sampling process three times and average the results over all trials to ensure comprehensive evaluations.

**Details of the graph construction.** Except for the illustration experiment in Figure 2, we construct the directed graph for all unlabeled data by connecting each vertex to its 10 nearest successors ($k = 10$). It is important to note that a larger $k$ will lead to an increase in the computation cost. We have chosen this setting because it provides good performance while maintaining efficient computation costs. For Figure 2, we construct the graph by connecting each vertex to its 3 nearest successors in order to avoid denseness.

**Details of Algorithm 1.** Considering the randomness of the diffusion process, when quantifying the influence of the subset, we run Algorithm 1 10 times and use the averaged influence value. Note that we also calculate the time cost in this repeated process when reporting the final results in the main paper. As shown in Figure 3, our algorithm is still more effective than Vote-$k$.

## E  TIME COMPLEXITY ANALYSIS

In this section, we perform a time-complexity analysis of our method. Given the constructed graph with $a$ nodes and $b$ edges, and an annotation budget of $m$, the whole algorithm involves the following two parts that incur the following time costs: (1) **Information diffusion process.** The time complexity of quantifying the influence of a specific subset is $\mathcal{O}(a+b)$. This is because it involves running an independent cascade diffusion process (BFS-like traversals) through the graph. (2) **Greedy search.** The greedy algorithm iteratively selects the example that provides the maximum marginal gain in influence. When the annotation budget is $m$, the time complexity is $\mathcal{O}(m*(a+b))$. This is because, in the worst case, the algorithm needs to evaluate the influence of each node in the network. Besides, for each node, the algorithm needs to perform a simulation with time complexity of $\mathcal{O}(a+b)$.

## F  LIMITATIONS

**Memory cost.** Although in-context learning tasks avoid the heavy parameter update process, they still require a large amount of memory to load models. For example, loading GPT-J 6B into a GPU requires about 23GB GPU memory, without considering the size of the dataset. This is a relatively high cost for individual researchers.

**The limitation of Auto-IDEAL.** Auto-IDEAL outperforms IDEAL in terms of performance, but it has two main drawbacks that hinder its usability in practice. First, Auto-IDEAL suffers from a model inference cost, especially in the era of large language models as a service. Specifically, when performing automatic annotation, Auto-IDEAL has to make predictions for all unlabeled examples to achieve automatic annotation. This makes it less practical than the native IDEAL. Second, Auto-IDEAL may fail when the initial examples to be labeled are not very relevant to the initial examples labeled, even though they have a similar embedding. Auto-IDEAL performs automatic annotation by following the information diffusion process from the initial annotated subset to the examples to be labeled with similar embedding. When the examples to be labeled are not relevant to the initial examples labeled, it may lead to incorrect automatic annotations and then poor performance. Future research may focus on maintaining superior performance while reducing the automatic annotation cost of IDEAL.

**Potential benchmark leakage in GPT-3.5-Turbo.** There may potentially exist benchmark leakage problems in the evaluation process (Zhou et al., 2023). Specifically, due to the fact that GPT-3.5-Turbo is trained using huge text datasets as of 2021, the data related to evaluation sets may potentially be used for model training. This could lead to potential risks in the evaluation process. However, as our work does not involve the training data selection, the impact should be negligible.

