# OpenReview forum: "IDEAL: Influence-Driven Selective Annotations Empower In-Context Learners in Large Language Models"
_ICLR.cc/2024/Conference — ICLR 2024 poster_

### Official Review · Reviewer_oY6L · 2023-10-31

**Soundness:** 3 good
**Presentation:** 3 good
**Contribution:** 3 good
**Rating:** 8
**Confidence:** 3

**Summary:**

In order to improve the performance of Large Language Models (LLMs) via in-context learning, this paper introduces a selective annotation method for creating, on a budget, high-value prompts for the LLM. The proposed approach selects a salient subset of examples to annotate (and choose-from for future prompts) from a large-scale unlabeled data pool. It uses the unlabeled data to (i) create a directed graph of "related examples," (ii) estimate  the influence of the candidates-for-annotation via a diffusion process, and (iii) iteratively select the data that provides a maximum marginal gain with respect to quantified influence.

**Strengths:**

The paper introduces a novel approach to improving in-context learning, which is an important problem with a myriad of strategic practical application. The works appears to be original and the. empirical evaluation suggests that this work could have major impact (also see comments below on how to make a stronger case along these lines). The paper is reasonably well-written and organized; see below suggestions on how to further improve it.

**Weaknesses:**

The most improvements could be achieved by tightening up the Empirical Evaluation section:
- ideally, rather than (arbitrarily?) choosing k = 18 & k = 100, you should provide an automated approach to AUTOMATICALLY assess the smallest amount of annotations that leads to the best possible performance; you only introduce Auto-IDEAL in "4.4" but you do not seem to claim it as a major contribution. Why?
- in Table 2:
   a) please add an additional row that shows the best-known performance on each dataset; without it, it is impossible to quantify the practical impact of IDEAL (i.e., is "just" improving on Vote-k, or does it come close to the best-known performance?)
   b) please also add to the Table results for k = 1,000 and even k = 10,000. How big are these improvements, if any?
   c) please explain why do we see IDEAL(100) < IDEAL(18) on RTE. Is there a general lesson worth sharing?
   d) similarly, why does Vote-k outperform IDEAL(100) on GeoQ (Tables 2 & 8). In particular, given that Table 8 suggest that there is a fair amount of variability among the various trials, ideally you should increase the number of trials to 10 or even larger. In the current setting, according to Table 8, for "100" Vote-k and IDEAL obtain the same best result (60.5) on GeoQ; similarly, for "18", there is a similar "tie" on HellaSwag, and Vote-k gets a better max on GeoQ.
- section 4.3.1 should cover all datasets, not only two; worst case scenario, if space is an issue, you can summarize the results on all datasets here, and refer the reader to the appropriate appendix
- same as above, section 4.3.1 should cover all datasets, not only three of them. Also: (i) why have Table 2 for "18" rather than "100," when IDEAL(100) outperforms IDEAL(1*0 on all datasets, and (2) similarly to the above comment on stability, you should use at least 10 randomized trials.
- same as above, section 4.4 should cover all datasets, not only five of them

OTHER COMMENTS:
1) please add early-on (e.g., in the Introduction) a few illustrative examples. For one or two of your application domains, you should show:
- an un-annotated example that IDEAL selects for annotation
- the annotated version of that example
- the answer of the LLM w/o that annotated example used as a prompt
- the answer of the LLM w/o that annotated example used as a prompt
- explain and/or discuss the difference between the two answers above

2) please add to the paper a paragraph or two about the actual/expected cost of annotation for the various domains in the empirical evaluation.

3) please intuitively explain the "end-to-end" term that you are using in the Abstract, Introduction, and later on in the Appendices, but nowhere else in the rest paper. If it is an important concept, it should also appear in the "meat" of the paper; if it is not, you should remove all references to it as it is distracting to see it mentioned early on, never clarified, and then simply ignored.

4) both in Fig 3's caption and in the abstract, please summarize the running time improvements of IDEAL vs Vote-k. The current "largely reduces the time cost" is simply too vague. Ideally, the last sentence of the abstract should end something like this: "... improving the performance by X% while reducing the running time by Y%"


Nitpicks:
- p 1: "pretty performance" --> "strong performance"
- p 1: please add a reference and /or brief summary for "requires substantial manpower and financial resources"
- p 2: "Vote-k devoids theoretical guarantees" --> "Vote-k lacks theoretical guarantees"
- p 2: please explain why "the algorithm does not need a delicate trade-off between diversity and representativeness"
- p 2: last paragraph of the Intro: please also quantify the performance improvement, not only the gains in speed
- p 3: for Fig 2, please explain if/why the iterations start at (d).
- p 5: probably the theorem should be called "Theorem 1" (or else the reader expects to see a Theorem 1 before it)

**Questions:**

1. why don't you claim Auto-IDEAL as a major contribution?
2. why don't you use all datasets in all experiments, but rather different subsets in 4.3.1, 4.3.2, and 4.4?
3. how did you end up with the two values of 18 and 10 (especially the "18" one)?
4. why do we see such large variability in performance among the three different runs? why din't you try more runs, so that you can get a better understanding of the scale of the variability issue?

---

> ### Author Response · Authors · 2023-11-16
> **Official Response to AnonReviewer oY6L - Part 1**
>
> We sincerely thank you for your careful reading of our paper and appreciate your valuable feedback in so many details.
>
> **1. [Re Weakness 1.1 - Part 1 & Question 3 :  Why choose 18 and 100 as the annotation budget?]**
>
> Thanks for your suggestion! We would like to clarify that the annotation budget is not arbitrarily selected but based on the following non-trivial reasons.
> - Firstly, we strictly followed the same setting as Vote-k, which also used 18 and 100 as the annotation budget, to enable easy comparison with the most important baseline and perform deep analysis.
> - Secondly, we included 18 as the annotation budget because all annotated examples can fit the prompt of the large language models within context limits. Therefore, the prompt retrieve stage can be ignored, and the evaluation results can naturally represent the quality of the selected examples.
>
> We appreciate your valuable suggestion and have revised the paper to make it clearer **in section 4.2**.
>
>
> **2. [Re Weaknesses 1.2 & Question 1: Why don’t include Auto-IDEAL as a major contribution.]**
>
> Although Auto-IDEAL has demonstrated better performance, it has two main drawbacks that may hinder its usability in practice. We list the reasons below.
>
> - Auto-IDEAL suffers from a model inference cost, especially in the era of large language models as a service. Specifically, when performing automatic annotation, Auto-IDEAL has to make predictions for all unlabeled examples to achieve automatic annotation. In our experiments on GeoQuery datasets, it costs more than 120 USD for a single run. Considering this, we think it is not as practical as the native IDEAL.
>
> - Auto-IDEAL may fail when the initial examples to be labeled are not very similar to the initial examples labeled, even though they have a similar embedding. Specifically, Auto-IDEAL performs automatic annotation by following the information diffusion process from the initial annotated subset to the examples to be labeled with similar embedding. When the examples to be labeled are not relevant to the initial examples labeled, it may lead to incorrect automatic annotation and poor performance.
>
> Considering these drawbacks, we decided to use native IDEAL as our core contribution but regarded Auto-IDEAL as a potential extension to IDEAL in a case study in our paper. We also illustrate it more clearly **in Appendix G** in the revised version.
>
> We may leave the potential improvement of Auto-IDEAL as our future work. Thank you for your suggestions!
>
> **3. [Re Weakness 1.1 - Part 2: You should provide an automated approach to automatically assess the smallest amount of annotations.]**
>
> Thank you for your valuable feedback! We argue that the concept of “the smallest amount of annotations” should not exist in selective annotation problems. Given that we do not know the distribution of the test examples, we believe that the larger the annotated examples, the better the expected performance, as it will be easier to retrieve the most relevant examples from a larger data pool. In an ideal case, all available examples should be annotated, and the algorithm could achieve the best-expected performance. However, in a more realistic setting, users have a limited budget, and the algorithm could select the best subset to annotate under the user’s budget. Regarding your suggestion of finding the smallest subset, we think a more meaningful setting would be if users have an expected performance threshold, and they expect there exists an algorithm that could help them find the smallest subset to annotate to reach the threshold

---

> ### Author Response · Authors · 2023-11-16
> **Official Response to AnonReviewer oY6L - Part 2**
>
> **4. [Re Weakness 2.1: Please also add an additional row to show the best known performance.]**
>
> Thanks for your suggestions. The main purpose of Table 2 is to compare the performance of IDEAL with other methods that could select a “coreset” from the large scale unlabeled data to annotate. Regarding the “best-known” performance, we can observe from Table 2 that IDEAL achieves the best performance compared with all baselines. If we misunderstand your suggestion, please let us know. Thanks!
>
> **5. [Re Weakness 2.2: Please also add the results when k=1000 even k=10000.]**
>
> Thank you for your suggestions. We would like to clarify that the value of K = 10000 exceeds the size of our dataset. To simulate the process of obtaining unlabeled samples, we followed the same setting as Vote-k, which is described on page 4 of Vote-k. In this setting, the sample size was set to 3000. Due to limited rebuttal time and computational resources, we will try our best to conduct additional experiments when k=1000 during the rebuttal period and update the results accordingly once complete as shown in the following reply. Thank you again for your suggestion!
>
>
> **6. [Re Weakness 2.3: please explain why we see IDEAL(100) < IDEAL(18) on RTE.]**
>
> Thank you for your suggestion. It is indeed an interesting scenario that was also observed in previous work Vote-k, where large annotation budgets do not necessarily lead to better performance (Vote-K (100) < Vote-K (18) in MWoZ, GeoQ, and DBpedia). However, in most cases (8 out of 9), IDEAL(100) performs better. We believe this particular scenario exists because a large annotated data pool may include some “bad” examples that have a similar embedding with the users’ query but are not relevant actually. Overall, a large annotation budget typically leads to better performance from a statistical perspective.
>
> **7. [Re Weakness 2.4: Why Vote-k outperform IDEAL(100) on GeoQ?]**
>
> Thank you for your constructive suggestion. Vote-k is insightful and needs a careful balance between diversity and representativeness. In some cases, e.g., on GeoQ (m=100), a good balance may be achieved. However, since the balance is tricky and complex, in most cases, Vote-k cannot work well, which motivates us to develop more advanced algorithms.
>
> We also kindly argue that the comparison of the two methods needs to be considered at the statistical level. IDEAL outperforms Vote-K in 17 out of 18 cases, which could indeed demonstrate the superiority of IDEAL over Vote-K. We totally agree that in an ideal case, as researchers, we should develop methods that are better than baselines in all cases. This is what we aim to achieve in our future research career and the follow-up work of our method. We appreciate your suggestions!
>
> **8. [Re Weakness 2.5 & Question 4: Why not include more trials in the experiments?]**
>
> Thank you for your suggestion. We included three different runs for each dataset for two reasons:
>
> - To ensure a fair comparison and ease of analysis, we used exactly the same setting as the most important baseline Vote-k, which also used three different runs.
>
> - Some experiments are especially costly. For example, each run of Mowz and GeoQ costs over 130+ USD dollars using GPT-3.5 Turbo. Considering our limited budget, we cannot afford large-scale experiments in some specific datasets.
> Due to limited rebuttal time and computational resources, we will try our best to conduct additional experiments to run more trials during the rebuttal period and update the results accordingly once complete, as shown in the following reply. Thank you again for your suggestion!

---

> > ### Comment · Reviewer_oY6L · 2023-11-18
> > **Thank you for the detailed answers!**
> >
> > First of all, thank you for taking the time to provide such detailed answers!
> >
> > I would like to rephrase my original comment below, which wasn't crisp enough to drive my point (apologies):
> >
> > >>>>>>>> in Table 2: a) please add an additional row that shows the best-known performance on each dataset; without it, it is impossible to quantify the practical impact of IDEAL (i.e., is "just" improving on Vote-k, or does it come close to the best-known performance?)
> >
> > It is great that IDEAL outperforms its competition, but, to judge its likely practical impact, you should also compare it against some sort of an upper-bound performance. To make a parallel with Active Learning, which has many similarities with your Selective Sampling approach, assume that you want to use 5 datasets D1-D5 to compare the accuracy of the current state-of-the-art  algorithm A1 against your newly-proposed approach A2, and you get the following table of results:
> >
> > TABLE-0        D1.       D2.       D3.       D4.       D5.
> > A1                  55%    80%     61%     32%     79%
> > A2                  65%    83%     69%     52%     83%
> >
> > Even though A2 clearly outperforms A1, there is a remaining question: is the performance of A2 good-enough for practical applications? If the datasets are hard, and the best-known performance when using all labeled examples is comparable to A2 (like in TABLE-1 below), the answer is YES
> >
> > TABLE-1        D1.       D2.       D3.       D4.       D5.
> > A1                 55%    80%     61%     32%     79%
> > A2                 65%    83%     69%     52%     83%
> > UseAllData    67%    84%     70%     52%    86%
> >
> > However, if the datasets are easy, like CIFAR-10 (see TABLE-2 below), the answer is NO. To stay a bit longer with the CIFAR analogy, an active learner with an accuracy of 82% is unlikely to be used in practice, when the best known performance on that dataset is 99%.
> >
> > TABLE-2        D1.       D2.       D3.       D4.       D5.
> > A1                 55%    80%     61%     32%     79%
> > A2                 65%    83%     69%     52%     83%
> > UseAllData    97%    94%     99%     97%     96%
> >
> > To summarize my point: if you can figure out a way to determine an upper-bound performance (naive as it may be) on your datasets, you will be able to make one two claims: either (1) IDEAL is a SOTA algorithm with massive practical applications, like in TABLE-1 above, or (2) IDEAL is better than Vote-k, but not yet ready to "move the needle" in the real world, like in TABLE-2 above. If you can compellingly make the point for "(1),"  the paper deserves a rating higher than I have currently assigned to it.

---

> > > ### Author Response · Authors · 2023-11-19
> > > **Official Response to AnonReviewer oY6L**
> > >
> > > **[Re Question:  Compare IDEAL against “upper-bound performance” to judge its likely practical impact]**
> > >
> > > We sincerely appreciate your insightful suggestions! We agree with your point that such a comparison is indeed meaningful.
> > >
> > > For selective annotation algorithms, the “upper bound performance” should be achieved when users have an unlimited annotation budget. In other words, all training data could be regarded as candidate In-context learning examples, and no selective annotation algorithm is performed. We constructed the “Oracle” algorithm, which uses all training data as candidate In-context learning examples for each dataset (3000 examples strictly following the same setting with Vote-K). Oracle still uses similarity-based example retrieval methods. We used the same experimental setting in Section 4.2 and presented the experimental results below.
> > >
> > > |           | MRPC | SST-5 | MNLI | DBpedia | RTE  | HellaSwag | MWoZ | GeoQ | Xsum |
> > > |-----------|------|-------|-----:|---------|------|-----------|------|------|------|
> > > | IDEAL-18  | 63.0 | 43.2  | 40.0 | 90.1    | 59.4 | 67.1      | 38.5 | 52.0 | 19.6 |
> > > | IDEAL-100 | 66.4 | 51.4  | **41.0** | 90.6    | 58.9 | 68.6      | 52.2 | 58.2 | 19.9 |
> > > | Oracle    | **69.5** | **53.2**  | 40.5 | **92.4**   | **60.2** | **70.7**      | **52.6** | **58.9** | **20.3** |
> > >
> > > We have made the following observations from our experiments:
> > >
> > > **Observation 1: Oracle outperforms both IDEAL-18 and IDEAL-100 in most cases.**
> > >
> > > Our experiments show that Oracle outperforms both IDEAL-18 and IDEAL-100 in most cases (8 out of 9 cases). This observation demonstrates that a larger annotation budget typically leads to better performance in general (Oracle (3000 examples) > IDEAL-100 > IDEAL-18). This result is predictable because it is more likely to retrieve similar In-context learning examples from a larger annotated data pool.
> > >
> > > **Observation 2: The performance difference between IDEAL-100 and Oracle is relatively small.**
> > >
> > > When considering selective annotation problems, we should take into account the practical impact of an algorithm from both performance and cost perspectives. Although larger annotation budgets typically lead to better performance, the annotation cost is huge (as we justified in Appendix F), and we cannot assume that users have an unlimited budget for any In-context learning tasks. According to the experiment results, although IDEAL-100 is worse than Oracle, it only uses about **3.3%** of the annotation budget of Oracle (100 examples vs. 3000 examples) with a slight performance sacrifice. This demonstrates the practical impact of the proposed method, especially considering that IDEAL is obviously better than other selective annotation baselines.
> > >
> > > **Observation 3: Some counterintuitive experimental results still occur (Oracle is worse than IDEAL-100 in MNLI).**
> > >
> > > As we illustrated in the previous response, a large annotated data pool cannot guarantee the best performance in all cases. However, a large annotation budget typically leads to better performance from a statistical perspective (8 out of 9 cases). When performing experiments on the MNLI dataset, the annotated data pool may include some “bad” examples that have a similar embedding with the query but are not relevant. Therefore, Oracle is slightly worse than IDEAL-100, which is a counterintuitive result.
> > >
> > > Overall speaking, compared with Oracle, IDEAL could save significant annotation cost with slight performance sacrifices in In-context learning tasks. We believe such an experiment could demonstrate the practical impact of the proposed selective annotation algorithm.
> > >
> > > Thanks again for your insightful comment and recognition of our work! If you have further questions, we will try our best to address them.

---

> ### Author Response · Authors · 2023-11-16
> **Official Response to AnonReviewer oY6L - Part 3**
>
> **9. [Re Question 2 & Weakness  4, 5 and Weakness 3 - part 1: Why don't you use all datasets in all experiments?]**
>
> Thank you for your insightful comments! We have the following three reasons for not performing experiments on all datasets for all sections in our paper, including ablations.
>
> (1) We intend to keep the same setting as previous work Vote-k for convenient comparisons and analysis, which also does not perform experiments on all datasets in the ablation sections. Moreover, to demonstrate the robustness of our method in different scenarios, we cover diverse datasets in different sections.
>
> (2) Performing selective annotation experiments is extremely costly in terms of both time and API fees. For example, it costs over 120 USD for a single run of GPT-3.5-turbo on Mowz and GeoQ datasets, which is unaffordable considering our limited budget.
>
> (3) We believe that using partial datasets in the main experiments to perform ablation is acceptable and does not result in a loss of generality. There are a large number of research works in both selective annotation [1] and more general In-Context learning examples selection [2][3][4] that only uses a partial dataset to perform the ablation experiments.
>
> We sincerely appreciate your suggestion, and we will try our best to perform experiments on all datasets in the ablation study during the rebuttal period, as shown in the following reply. Thank you again for your suggestion!
>
> [1] Su, H., Kasai, J., Wu, C. H., Shi, W., Wang, T., Xin, J., ... & Yu, T. (2022). Selective annotation makes language models better few-shot learners. arXiv preprint arXiv:2209.01975.
>
> [2] Chang, T. Y., & Jia, R. (2023, July). Data curation alone can stabilize in-context learning. In Proceedings of the 61st Annual Meeting of the Association for Computational Linguistics (Volume 1: Long Papers) (pp. 8123-8144).
>
> [3] Nguyen, T., & Wong, E. (2023). In-context Example Selection with Influences. arXiv preprint arXiv:2302.11042.
>
> [4]Li, X., & Qiu, X. (2023). Finding supporting examples for in-context learning. arXiv preprint arXiv:2302.13539.
>
> **10. [Re Weakness 3 - part 2: Why have Table 2 for "18" rather than "100”?]**
>
> Thanks for your suggestions. In most cases, we choose to use 18 as the annotation budget. This is because all annotated examples can fit the prompt of the large language models within context limits. Therefore, the prompt retrieval stage is ignored and the evaluation results can naturally represent the quality of the selected examples. In section 4.3.2, the main purpose of the experiment is to demonstrate that the selected annotated examples could work well in multiple LLMs. To exclude the influence of other factors, we chose 18 as the annotation budget.
>
> **11. [Re Comments 1:Adding a few illustrative examples in the introduction?]**
>
> We sincerely appreciate your suggestion. Due to space constraints, we would like to clarify that the performance improvement brought about by the annotated examples, as compared to the performance without annotated examples (zero-shot), is due to the few-shot learning ability of LLMs itself [1][2], rather than our selection annotation method. Therefore, we have not included such illustrative examples in our paper. However, we have provided some selected examples in **Appendix C** to demonstrate the core example selection ability of our algorithm, which is the primary contribution of our method.
>
> [1] Brown, T., Mann, B., Ryder, N., Subbiah, M., Kaplan, J. D., Dhariwal, P., ... & Amodei, D. (2020). Language models are few-shot learners. Advances in neural information processing systems, 33, 1877-1901.
>
> [2] Wei, J., Wang, X., Schuurmans, D., Bosma, M., Xia, F., Chi, E., ... & Zhou, D. (2022). Chain-of-thought prompting elicits reasoning in large language models. Advances in Neural Information Processing Systems, 35, 24824-24837.
>
> **12. [Re Comments 2: please add to the paper a paragraph or two about the actual/expected cost of annotation for the various domains in the empirical evaluation?]**
>
> In our revised version, we have added one section in the Appendix to further illustrate the cost of annotation (Appendix F) . Thanks for your insightful suggestion.

---

> ### Author Response · Authors · 2023-11-16
> **Official Response to AnonReviewer oY6L - Part 4**
>
> **13. [Re Comments 3: Please intuitively explain the "end-to-end" term that you are using in the Abstract, Introduction, and later on in the Appendices.]**
>
> Thank you for your feedback. We refer to IDEAL as an end-to-end algorithm because it directly selects a subset of unlabeled data for annotation from a large-scale pool in one step, without any intermediate procedure. In contrast, Vote-k selects a subset of data in two stages. First, it selects a small portion of data for diversity and annotates them manually. Second, these annotated data act as prompts for predictions on all other unlabeled data, and the remaining ones that need to be annotated are chosen based on diverse confidence scores. This approach adds inference costs due to the predictions on unlabeled data. We mentioned the difference between IDEAL and Vote-k in our original submission.
>
> **14. [Re Comments 4:  please summarize the running time improvements of IDEAL vs  Vote-k?]**
>
> Thank you for your feedback. In our original submission, we summarized the numerical improvement with respect to time consumption on page 7. “Specifically, under the same hardware conditions, IDEAL achieves a 7.8× lead on average over Vote-k.”. In response to your constructive suggestion, we also have added this information to the caption to further emphasize the advantage of our method in the revised version. Thank you again for your suggestions!
>
> **15. [Re Nitpicks: P1 - P5]**
>
> Thanks for your suggestions. We have revised the paper according to your suggestions from P1, P2, P4, P5 which is shown in blue color. For P3, we would like to clarify that the iteration doesn’t start from (d).
> We deeply appreciate the time and effort you spent to further improve our submission. Thanks!
>
>
> -------------------------------------- **Experiments Work in Progress** --------------------------------------
>
>
> Due to the extremely limited computation resources and rebuttal time, we will do our best to add as many experiments as possible during the rebuttal period and update the results accordingly. While we cannot guarantee that we will be able to complete all the experiments in a short period of time, we will make every effort to do so. We will also update the results in our future version of the paper. Thank you for your valuable suggestions!
>
> —------------------------------ **Update : Nov 16 2023** —------------------------------
>
> **Completed**
>
> Five seeds results for Table 4.2 with annotation budget 18 (Will run five more seeds)
>
> |        | MRPC | SST-5 | MNLI | DBpedia | RTE  | HellaSwag | MWoZ | GeoQ | Xsum |
> |--------|------|------:|------|---------|------|-----------|------|------|------|
> | Random | 56.4 |  42.1 | 36.8 | 84.3    | 56.0 | 64.5      | 37.6 | 46.8 | 14.9 |
> | Vote-K | 62.5 |  41.1 | 38.8 | 88.5    | 57.4 | 65.9      | 38.2 | 50.2 | 16.7 |
> | IDEAL  | **64.3** |  **43.8** | **39.8** | **90.8**    | **58.8** | **66.8**      | **39.9** | **52.5** | **19.1** |
>
>
> **Current In progress**
>
> Using all datasets to perform experiments on section 4.3.1

---

> > ### Author Response · Authors · 2023-11-21
> > **Response to AnonReviewer oY6L**
> >
> > —------------------------------ **Update : Nov 19 2023** —------------------------------
> >
> > **Completed**
> >
> > We got additional results for Section 4.3.1 on the remaining classification tasks (MRPC, DBpedia, RTE) and presented the results below. The same setting as Section 4.3.1 is strictly followed. We still observe that larger influence lead to better performance.
> >
> > |                    | MRPC | DBpedia | RTE  |
> > |--------------------|------|---------|------|
> > | Influence Level 1  | 54.3 | 81.6    | 53.2 |
> > | Influence Level 2  | 55.8 | 84.0    | 54.4 |
> > | Influence Level 3  | **57.9** | **85.5**    | **56.7** |
> >
> > **Current In progress**
> >
> > Ten different runs for the main experiments.

---

> > > ### Author Response · Authors · 2023-11-21
> > > **Response to AnonReviewer oY6L**
> > >
> > > We sincerely appreciate your effort for reviewing our paper. We got more experimental results as shown below.
> > >
> > > —------------------------------ Update : Nov 21 2023 —------------------------------
> > >
> > > Upon to Nov 21 2023, we got the ten seeds results for Table 4.2 with annotation budget 18 and show the results below. We can observe that IDEAL is still better than baselines demonstrating its stable performance in practice.
> > >
> > > |        | MRPC | SST-5 | MNLI | DBpedia | RTE  | HellaSwag | MWoZ | GeoQ | Xsum |
> > > |--------|------|------:|------|---------|------|-----------|------|------|------|
> > > | Random | 58.4 |  40.5 | 35.9 | 83.3    | 57.9 | 66.7      | 36.6 | 48.8 | 13.5 |
> > > | Vote-K | 61.5 |  41.5 | 37.1 | 88.3    | 58.1 | 66.9      | 37.2 | 50.2 | 15.8 |
> > > | IDEAL  | **62.8** |  **42.8** | **39.1** | **89.8**    | **59.1** | **67.8**      | **38.7** | **51.4** | **19.3** |
> > >
> > > Current In progress
> > >
> > > Ten different runs for the main experiments with annotation budget 100.

---

> > > > ### Author Response · Authors · 2023-11-22
> > > > **Thanks for your effort**
> > > >
> > > > Dear Reviewer oY6L:
> > > >
> > > > Thanks a lot for your efforts in reviewing this paper! We tried our best to address the mentioned concerns and have provided a detailed response. Could you please kindly re-evaluate our paper based on the current situation?
> > > >
> > > > As the discussion deadline is very close, we authors want to confirm whether there are unclear explanations and descriptions here. We could further clarify them.
> > > >
> > > > Thanks!
> > > >
> > > > Authors

---

> > > > ### Author Response · Authors · 2023-11-22
> > > > **Response to AnonReviewer oY6L**
> > > >
> > > > We sincerely appreciate your effort in reviewing our paper. We got more experimental results as shown below.
> > > >
> > > > —------------------------------ Update : Nov 22 2023 —------------------------------
> > > >
> > > > Upon to Nov 22 2023, we got the five seeds results for Table 4.2 with annotation budget 100 and show the results below. We can observe that IDEAL is still better than baselines.
> > > >
> > > > |        | MRPC | SST-5 | MNLI | DBpedia | RTE  | HellaSwag | MWoZ | GeoQ | Xsum |
> > > > |--------|------|------:|------|---------|------|-----------|------|------|------|
> > > > | Random | 61.8 |  48.7 | 36.9 | 86.4    | 55.8 | 66.1      | 40.2 | 54.8 | 15.7 |
> > > > | Vote-K | 63.5 |  47.5 | 38.1 | 88.8    | 58.2 | 68.1      | 46.2 | 56.7 | 18.8 |
> > > > | IDEAL  | **65.8** |  **50.8** | **38.8** | **89.4**    | **59.1** | **68.8**      | **50.5** | **57.5** | **19.2** |
> > > >
> > > > Current In progress
> > > >
> > > > five more different runs for the main experiments with an annotation budget 100.

---

> > > > > ### Author Response · Authors · 2023-11-23
> > > > > **Seeking Confirmation for Resolved Concern and Final Rating with gratitude**
> > > > >
> > > > > Dear Reviewer oY6L,
> > > > >
> > > > > Thanks a lot for your efforts in reviewing this paper. We tried our best to address the mentioned concerns particularly regarding the comparisons between our method and the Oracle method.
> > > > >
> > > > > Could you please kindly re-evaluate our paper based on the current situation? If you have any further questions, we are also very glad to discuss them. We appreciate your careful review.
> > > > >
> > > > > Best,
> > > > >
> > > > > Authors

---

### Official Review · Reviewer_QkNp · 2023-11-01

**Soundness:** 2 fair
**Presentation:** 3 good
**Contribution:** 3 good
**Rating:** 6
**Confidence:** 3

**Summary:**

This paper studies the selective annotation for minimizing annotation cost on massive unlabeled in-context examples. To tackle the drawbacks of the existing Vote-k method and choose diverse and representative in-context examples, this paper proposes an unsupervised influence diffusion method to quantify the influence of each unlabeled example in the example corpus. Experimental results on various datasets and LLMs validates its effectiveness with reduced time consumption. The paper also proves the influence lower bound of the in-context selection method.

**Strengths:**

- The selective annotation is a meaningful research problem, since in reality LLMs need to to generalize to novel tasks without the existence of large amount of labeled data.
- The previous Vote-k method needs to get the uncertainty from LLM predictions which incurs large computation time consumption, while the proposed method bypasses such need and considers from the perspective of data influence in the corpus. Also, this method reduces the LLM sensitivity on in-context examples, such as the order, which increases the robustness of in-context learning.
- The experiments is conducted on different LLMs and various tasks, showing its effectiveness.

**Weaknesses:**

- This paper is related to "Li et al., Finding Support Examples for In-Context Learning", which also adopts the idea of coreset selection in in-context example selection, and "Diao et al., Active prompting with chain-of-thought for large language models", It is suggested to include discussion with these papers in the related work.
- The theoretical analysis is about the influence lower bound. How the influence score relates to better diversity and representativeness needs further illustration, since it is the major consideration of the proposed method.
- A minor suggestion is that the experiment datasets can also include some novel LLM benchmarks that are probably not trained on the experiment LLMs.

**Questions:**

- This paper uses small LMs less than 7B, and ChatGPT. Will the method still be effective on medium-sized LMs such as LLaMA2-chat 7B, 13B? And do you use any instruction for testing on ChatGPT?

---

> ### Author Response · Authors · 2023-11-16
> **Official Response to AnonReviewer QkNp**
>
> **1. [Re Weaknesses 1: Include discussion with an important baseline in the main paper.]**
>
> We appreciate the suggestions. We have revised the paper according to the suggestion and include the discussion of these two papers **in the related work section in the revised version**.  Thanks for your valuable feedback!
>
> **2. [Re Weaknesses 2: How the influence score relates to better diversity and representativeness.]**
>
> Thank you for your feedback! We would like to clarify that our method is not intended to select examples to achieve better diversity and representativeness. Instead, we aim to investigate a new paradigm to quantify potential selective annotation subsets through an information diffusion process that avoids the intractable explicit balance between data diversity and representativeness, as mentioned in our paper’s abstract. Furthermore, we provide a visualization for the selected examples in Figure 6 in our paper that compares Vote-$k$ and IDEAL. Vote-$k$ takes diversity into consideration, which may lead to the inclusion of outliers.
>
> **3. [Re Weaknesses 3: Include experiment datasets that are probably not trained on the experiment LLMs.]**
>
> Thank you for your constructive suggestions! We agree that benchmark leakage is a significant research problem that has garnered a lot of attention in the research community [1]. However, the research problem in our paper is selective annotation in In-Context learning, which does not involve the processing of training data selection. Due to the limited rebuttal time, we will take a deep dive into the benchmark leakage problem in In-Context Learning in our future research and investigate its impact on selective annotation problems. We have also added an additional paragraph in **Appendix G** to discuss the benchmark leakage problems. Thank you for your suggestions!
>
> [1] Zhou, K., Zhu, Y., Chen, Z., Chen, W., Zhao, W. X., Chen, X., ... & Han, J. (2023). Don't Make Your LLM an Evaluation Benchmark Cheater. arXiv preprint arXiv:2311.01964.
>
> **4. [Re Question 1: Will the method still be effective on medium-sized LLMs such as LLaMA2-chat 7B/13B?]**
>
> Thank you for your valuable suggestions. We conducted experiments and evaluated the results of our method and baselines on LLaMA2-7b-chat and LLAma2-13b-chat using the same setting as Section 4.3.4. Our observations indicate that IDEAL consistently outperforms the baselines. This demonstrates that our method is a general approach that works well in medium-sized LLMs.
>
> **---------------------------- LLaMA2-7b-chat ----------------------------**
>
>
> |        | MRPC | MNLI | RTE  |
> |--------|------|------|------|
> | Random | 64.6 | 37.9 | 56.3 |
> | Vote-K | 66.0 | 39.1 | 58.6 |
> | IDEAL  | **67.2** | **39.4** | **60.5** |
>
> **---------------------------- LLaMA2-13b-chat ----------------------------**
>
> |        | MRPC | MNLI | RTE  |
> |--------|------|------|------|
> | Random | 65.8 | 38.8 | 58.5 |
> | Vote-K | 66.5 | 40.1 | 59.2 |
> | IDEAL  | **66.8** | **40.9** | **60.7** |
>
> **5. [Re Question 2: Do you use any instruction on ChatGPT?]**
>
> Thank you for your feedback. To ensure a fair and easy comparison, we used the same instructions and prompts as the other LLMs used in our experiments. This approach allows us to more strictly show the differences between multiple LLMs and verify the generalization ability of our method on different LLMs. We have elaborated on this procedure in more detail in our revised version (**Section 4.3.4**).

---

> > ### Comment · Reviewer_QkNp · 2023-11-19
> >
> > I don't have further questions and would like to keep my rating on the positive side.

---

> > > ### Author Response · Authors · 2023-11-19
> > > **Response to Reviewer QkNp**
> > >
> > > Thanks for your suggestions and your valuable feedback. Your comment is very meaningful to further improve our manuscript!

---

### Official Review · Reviewer_Ra1D · 2023-11-01

**Soundness:** 3 good
**Presentation:** 3 good
**Contribution:** 3 good
**Rating:** 3
**Confidence:** 4

**Summary:**

In this paper, the authors propose an influence-driven demonstration selection method. I like the theoretical analysis of the influence function. However, I hold two main concerns: (i) if it requires running the influence function during the inference, then it is crucial to have the time complexity analysis, (ii) how the proposed approach considers the correlations among the retrieved data points.

**Strengths:**

1. Investigating the demonstration selection task from the influence perspective is very interesting.
2. It is great to have a theoretical analysis of the property of the influence function.
3. Overall, this paper is well written.

**Weaknesses:**

1. There is no time complexity in the proposed method, which is very crucial if it needs to run for every inference.
2. It is not clear how the proposed method considers the correlations among the retrieved data points.
3. It is better to compare with more demonstration selection methods such as similarity-based and diversity-based methods which are widely used in practice.

**Questions:**

The topic of the demonstration selection is a hot and essential topic. I like the idea of investigating the demonstration selection from the perspective of influence. I like Section 3. I have two main concerns. First is that for each inference of LLM, whether it is required to run the influence function for n times to retrieve n shots. If it is true, it is very important to have a complexity analysis of running time. Second, I do not see how to consider the correlations among the retrieved samples. For greed search, there may be cases where A+B is better C+B, and C has more influence than A. I want to know how the proposed method addresses these cases. Furthermore, I also want to know how the proposed method leverages the features of each data sample. In other words, the nodes in the graph have their own features. How do you embed these features? Using sentence transformer or some GNN-based methods? And, what is the cost of applying these methods? Also, for the experiment, there are many demonstration selection methods in the NLP field, for example, similarity-based or diversity-based methods, which are widely employed in the LLMs. Therefore, I highly recommend the authors compare their method against these existing approaches. It would be great to see some results for more datasets (especially more tasks beyond the text classification) and more LLMs. Overall, I do not think this version is ready for publication.

---

> ### Author Response · Authors · 2023-11-16
> **Official Response to AnonReviewer Ra1D - Part 1**
>
> Thanks for your constructive suggestions. The insightful and constructive suggestions have enabled us to effectively improve our work. Please find our response in the following text.
>
> **1. [Re  weakness 1 & question 1: Whether it is required to run the influence function for n times to retrieve n shots?]**
>
> Thank you for your constructive suggestion. To retrieve n shots, the influence function indeed needs to be run n times. We also included a time-complexity analysis **in the Appendix E** in the revised version. Here is a high-level overview of the time-complexity analysis for our IDEAL.
>
> - Information diffusion process: The time complexity of quantifying the influence of a specific subset is O(a + b), where a is the number of edges and b is the number of nodes. This is because it involves running an independent cascade diffusion process (BFS-like traversals) through the graph.
>
> - Greedy algorithm: The greedy algorithm iteratively selects the example that provides the maximum marginal gain in influence. When the annotation budget is n, the time complexity is O(n*(a+b)). This is because, in the worst case, the algorithm needs to evaluate the influence for each of the n nodes in the network, and for each node, the algorithm needs to perform a simulation with a time complexity of O(m + n).
>
> Regarding time consumption, we compared IDEAL with Vote-$k$ in Figure 3 in the log scale. We observed that IDEAL achieves a **7.8× lead** on average over Vote-k. This time consumption improvement benefits from the fact that IDEAL doesn’t involve any model prediction time cost, which is totally unsupervised compared to Vote-$k$. Therefore, the proposed method is more efficient.
>
> **2. [Re weakness 2 & question 2: How to consider the correlations among the selected examples when performing greedy search?]**
>
> We appreciate your suggestion and address your concerns as follows.
> - Our method does not require consideration of correlations among retrieved data points. In fact, this is precisely the advantage of our method. IDEAL aims to maximize the influence of the entire selected subset, rather than considering each example separately.
> - Taking the example you provided, if the influence of A+B is greater than C+B, our algorithm would choose A+B over C+B, even though C has a greater influence than A. As mentioned in section 2.2, IDEAL aims to maximize the influence of the entire selected subset of examples, which avoids an intractable explicit balance between data diversity and representativeness.
>
> Additionally, we also have supplemented more details in the revised paper **(section 2.2)** based on your advice. Thanks for this suggestion!
>
> **3. [Re weakness 3 & question 4 : Compare with more demonstration selection methods in NLP fields including similarity-based or diversity-based.]**
>
> Thanks for the suggestion. We address your concern below.
> - We kindly argue that the research problem in our paper is the selective annotation problem, which involves curating a core subset from a large-scale data pool to annotate. The “similarity-based or diversity-based” methods you mentioned are demonstrated retrieval methods that are not what we investigated in this research. Our method is agnostic to example retrieval methods and can be combined with all potential retrieval methods.
> - In particular, we also investigated different retrieval methods **in Section 4.3.3** when combined with IDEAL. We observed that similarity-based methods achieve better performance.

---

> ### Author Response · Authors · 2023-11-16
> **Official Response to AnonReviewer Ra1D - Part 2**
>
> **4. [Re question 3: How to get the embedding of the examples when constructing the graph? And what is the cost?]**
>
> As we illustrated in our paper **(page 3 in our submission)**, we obtain the embedding of the examples through Sentence-BERT which follows the same setting as Vote-$k$. The cost of this process is negligible and can be ignored, as it takes less than 5 minutes for all datasets in our experiments.
>
> **5. [Re question 5:  It would be great to see some results for more datasets (especially more tasks beyond the text classification) and more LLMs.]**
>
> Thank you for your constructive feedback. We address your question below.
>
> - Following the same setting with previous selective annotation work Vote-K in the same area, we have demonstrated the results of IDEAL for datasets beyond text classification, including Multi-Choice, Dialogue, and Generation in **Section 4.2**. We have also conducted experiments for different LLMs **in section 4.3.4**.
> - Additionally, we have performed additional experiments on medium-sized LLMs LLaMA2-chat 7B and 13B using the same setting as Section 4.3.4, and presented the results below. Our observations indicate that IDEAL consistently outperforms the baselines. This demonstrates that our method is a general approach that works well in medium-sized LLMs
>
> **------------------------------ LLaMA2-7b-chat ------------------------------**
>
> |        | MRPC | MNLI | RTE  |
> |--------|------|------|------|
> | Random | 64.6 | 37.9 | 56.3 |
> | Vote-K | 66.0 | 39.1 | 58.6 |
> | IDEAL  | **67.2** | **39.4** | **60.5** |
>
> **------------------------------ LLaMA2-13b-chat ------------------------------**
>
> |        | MRPC | MNLI | RTE  |
> |--------|------|------|------|
> | Random | 65.8 | 38.8 | 58.5 |
> | Vote-K | 66.5 | 40.1 | 59.2 |
> | IDEAL  | **66.8** | **40.9** | **60.7** |

---

> ### Author Response · Authors · 2023-11-19
> **Looking forward to your reply**
>
> Dear Reviewer Ra1D:
>
> Thanks again for your efforts in reviewing. We are looking forward to your reply. Would you mind checking our response and confirming if there are unclear explanations?
>
> We are highly encouraged if your concerns have been addressed. On the contrary, if you need any more clarification, we can provide it as soon as possible before the discussion deadline.
>
> Thanks!
> Authors

---

> ### Author Response · Authors · 2023-11-20
> **Thanks for your effort!**
>
> Dear Reviewer Ra1D:
>
> Thanks a lot for your efforts in reviewing this paper. We tried our best to address the mentioned concerns and have provided a detailed response. Are there unclear explanations here? We could further clarify them.
>
> Thanks!
>
> Authors

---

> ### Author Response · Authors · 2023-11-21
> **Looking forward to your reply!**
>
> Dear Reviewer Ra1D:
>
> Thanks a lot for your efforts in reviewing this paper! We tried our best to address the mentioned concerns and have provided a detailed response.
>
> As the discussion deadline is very close, we authors want to confirm whether there are unclear explanations and descriptions here. We could further clarify them.
>
> Thanks!
>
> Authors

---

### Official Review · Reviewer_Ctph · 2023-11-05

**Soundness:** 3 good
**Presentation:** 3 good
**Contribution:** 3 good
**Rating:** 5
**Confidence:** 4

**Summary:**

In this paper, the authors propose the method using influence-driven selective annotations to minimize the annotation cost, thereby tackling the high annotation costs of discovering the right prompts for in-context learning. They construct a direct graph to represent unlabeled data and use a diffusion process to quantify the influence of unlabeled subsets. Finally, a greedy algorithm is utilized to conduct the final selection. Experiments are conducted on several benchmark datasets, and the experimental results show that the proposed method can achieve better performance with lower computational time in subset selection.

**Strengths:**

* S1: The influence-based greedy method can effectively and efficiently help find the right prompts from the large corpus.
* S2: Experimental results show improvements over Random and Vote-K.
* S3: The authors provide some preliminary theoretical proofs.

**Weaknesses:**

* W1: Some state-of-the-art studies like MDL [a] TopK (Liu et al., 2022) are not cited and/or compared in the experiments.
* W2: The rationales behind some experimental results are not explained, such as the reverse performance of MNLI in Table 5.
* W3: Lack of analysis on how the proposed method



[a] Wu, Z., Wang, Y., Ye, J., & Kong, L. Self-Adaptive In-Context Learning: An Information Compression Perspective for In-Context Example Selection and Ordering, ACL 2023.

**Questions:**

* Q1: Following W1,it would be great if the authors could patch more comparisons to more state-of-the-art works during the author feedback period.
* Q2: Following W2, I wonder why there are some inconsistent behaviors of the proposed method, instead of only getting the message "the proposed method performs still not bad". This is important because it would decide the choice between IDEAL and Auto-IDEAL.
* Q3: Following W3, because the proposed method should be general, it is reasonable to apply the method to different LLMs. It would be great if the authors could also conduct some related studies.

**Details Of Ethics Concerns:**

N/A.

---

> ### Author Response · Authors · 2023-11-16
> **Official Response to AnonReviewer Ctph**
>
> We sincerely thank you for your careful reading of our paper and appreciate the valuable feedback in your comments. The insightful and constructive suggestions have enabled us to effectively improve our work.
>
> **1. [Re Weakness 1 & Question 1:  Comparisons with Self-adaptive in context learning and TopK.]**
>
> Thank you for your constructive feedback! In our revised version, we have included a detailed discussion **in the related work section (Appendix A)** for Wu et al., 2022 and TopK (Liu et al. 2022 and Gao et al).
>
> We kindly clarify that the research problem in our paper is different from these papers and the methods are also agnostic. Specifically,
> Wu et al., 2023, is the first work to formally define the problem of self-adaptive in-context learning, which aims to search for the best in-context example and corresponding order with a set of annotated candidate examples.
> Liu et al. 2022 and Gao et al. 2021  focus on the example retrieval problem which proposes to use the nearest neighbors of a given test query as the corresponding In-Context examples.
>
> On the contrary, the research problem of our paper is the selective annotation problem, which aims to curate a core set from large-scale unlabeled data to annotate. This problem is agnostic to the both self-adaptive In-Context Learning problem and example retrieval.
>
> Additionally, we also provide ablation experiments in Section 4.3.3 to quantify the effect of different example retrieval methods. We believe that all three research problems are meaningful and worth exploring in the research community. Thanks again for your suggestion!
>
> **2. [Re Weakness 2 & Question 2: The rationales behind the reverse performance of MNLI in Table 5.]**
>
> We sincerely appreciate your constructive suggestions. We address your concerns as follows.
>
> - The performance drop of MNLI in Table 5 is predictable because Auto-IDEAL is an automatic annotation method that relies on half of the selected examples being automatically annotated rather than manually labeled ground truth.
>
> - Auto-IDEAL performs automatic annotation by following the information diffusion process from the initial annotated subset to the examples to be labeled. In some datasets, when the examples to be labeled are not very relevant to the initial examples labeled (but with a similar embedding), it may lead to a performance drop. We acknowledge that this is a potential drawback of Auto-IDEAL. Therefore, we did not regard Auto-IDEAL as the main contribution of our paper but chose a more trivial method IDEAL.
>
> We appreciate your suggestions and have elaborated more specifically on this issue in the revised versions in **Appendix G.**
>
> **3. [Re Question 3 & Weakness 3: It would be great to apply IDEAL to different LLMs.]**
>
> Thank you for your constructive suggestions!
> - We totally agree that IDEAL is a general selective annotation method. We would like to clarify that we have shown the evaluation results of different methods using multiple LLMs **in Figure 5** in our submission. Our evaluations reveal that IDEAL consistently outperforms the baselines across all models tested.
>
> - Additionally, we further performed more experiments on medium-sized LLMs LLaMA2-7B-chat, LLaMA2-13B-chat using the same setting as Section 4.3.4, and showed the results below. We observed that IDEAL still achieves the best performance compared to the baselines.
>
> **------------------------------ LLaMA2-7b-chat ------------------------------**
>
> |        | MRPC | MNLI | RTE  |
> |--------|------|------|------|
> | Random | 64.6 | 37.9 | 56.3 |
> | Vote-K | 66.0 | 39.1 | 58.6 |
> | IDEAL  | **67.2** | **39.4** | **60.5** |
>
> **------------------------------ LLaMA2-13b-chat ------------------------------**
>
> |        | MRPC | MNLI | RTE  |
> |--------|------|------|------|
> | Random | 65.8 | 38.8 | 58.5 |
> | Vote-K | 66.5 | 40.1 | 59.2 |
> | IDEAL  | **66.8** | **40.9** | **60.7** |

---

> ### Author Response · Authors · 2023-11-19
> **Thanks for your effort!**
>
> Dear Reviewer Ctph
>
> We have properly addressed all your concerns and provided clarifications on all confusing concepts. Could you please kindly re-evaluate our paper based on the current situation? If you have any further questions, we are also very glad to discuss them.
>
> Thanks!
>
> Authors

---

> ### Author Response · Authors · 2023-11-20
> **Looking forward to your reply**
>
> Dear Reviewer Ctph:
>
> We sincerely appreciate your efforts in reviewing this paper! We tried our best to address the mentioned concerns about related work, experimental results, and more empirical evidence. The discussion deadline is approaching now. Are there unclear explanations and descriptions here?
>
> We are highly encouraged if your concerns have been addressed. On the contrary, if you need any more clarification, we can provide it as soon as possible before the discussion deadline.
>
> Thanks!
>
> Authors

---

> ### Author Response · Authors · 2023-11-21
> **Further Discussion**
>
> Dear Reviewer Ctph:
>
> Thanks a lot for your efforts in reviewing this paper. We tried our best to address the mentioned concerns. As the discussion deadline between reviewers and authors is very close, we tend to confirm whether there are unclear explanations and descriptions here. We could further clarify them.
>
> Thanks!
>
> Authors

---

### Official Review · Reviewer_RM4K · 2023-11-07

**Soundness:** 3 good
**Presentation:** 4 excellent
**Contribution:** 3 good
**Rating:** 8
**Confidence:** 5

**Summary:**

This paper presents a work on selective annotations in in-context learning. Given a pool of unlabeled instances to annotate for prompts, this paper proposes an influence-driven framework to identify those more significant data points for annotations and follow-up tasks. Compared with the previous method, the proposed method enjoys (1) greater convenience in selection due to the end-to-end manner; (2) it does not need to balance diversity and representativeness of selected instances; (3) theoretical analysis is provided to show the subset influence is at least as large as a certain proportion of the influence of the optimal solution. Comprehensive experiments are conducted to show the superiority of the proposed method, which achieves the best performance in most cases. The time consumption in selection is significantly lower than previous state-of-the-art methods.

**Strengths:**

1.	In-context learning is an important and swiftly advancing research area, capturing significant attention from both the ICLR and NLP communities. This paper conducts comprehensive evaluations on the influence of selecting specific examples for inclusion in the prompt. The findings illustrate the substantial impact this choice can have on the final model's behavior.
2.	The paper is well-structured and easily comprehensible. Within the evaluation section, there are numerous intriguing findings that hold significant value for dissemination within the wider research community. Besides, the source code is provided to ensure the reproducibility of the empirical findings.
3.	The proposed method is simple and relatively straightforward to implement. Consequently, there is a possibility that it could be utilized in real-world applications of in-context learning.
4.	Theoretical analysis is provided to demonstrate the effectiveness of the proposed method under a greedy search algorithm.

**Weaknesses:**

1.	There are some unclear expressions and inconsistent explanations. Some polishes are needed.
2.	Some experimental settings only include the baselines Random and Vote-k. More methods as mentioned in 4.3.2 can be also included.

More questions about weaknesses can be checked below.

**Questions:**

1)	The experiments can be supplemented by including other methods not just Random and Vote-k, as mentioned above.
2)	The paper claims that it conducts experiments three times and reports the average score. What/where is the randomness? It mainly comes from examples selected or model predictions? Or both of them?
3)	For the process of information diffusion, in the main paper, it seems that the process is only performed once. In the Appendix, the paper discusses that multiple processes are performed. I suggest that this description can be moved to the main paper. Also, what is the influence of the times of information diffusion?
4)	Additional elaboration is required to explain why the combination of IDEAL/vote-k and similarity retrieval proves effective, while IDEAL/Vote-k alone, coupled with the random selection of supporting examples, does not work well.
5)	Recently, Chain-of-Thought (CoT) prompting has gained widespread adoption in in-context learning research. Several studies employing this method have reported enhanced model performance. However, an unexplored aspect of this approach is the model's sensitivity and variance concerning the retrieved prompts when used in conjunction with CoT. This work does not address whether such sensitivity exists and whether CoT can be applied for better performance.
6)	Is there any consideration given by the authors to the cost associated with encoding every unlabeled training instance using Sentence-BERT when comparing the methods? As far as I can discern, this aspect was not addressed in the paper.
7)	It seems that the instance embedding has a large Influence on the proposed method since it needs to build a directed graph using the embeddings. I am interested in changing the previously used pre-trained models and seeing the robustness of the proposed method.
8)	The paper shows the label distribution brought by the proposed method. What about the distribution of original data (before selecting)? Are their label distributions similar? Besides, a balanced distribution is expected to bring better performance in follow-up tasks? Or otherwise. I am interested in this.

---

> ### Author Response · Authors · 2023-11-16
> **Official Response to AnonReviewer RM4K**
>
> We sincerely thank the reviewer for the insightful comments. Please find our response to your comments below.
>
> **1. [Re Question 1 & Weakness 2: Should the experiment results of other methods be included in the paper?]**
>
> Thank you for your valuable feedback. We would like to clarify that the research problem in this paper is a selective annotation problem and we have compared our method with alternative selective annotation methods, excluding vote-k and random search in **Section 4.3.2.** The evaluation results show that IDEAL outperforms all alternative methods, which could demonstrate the superiority of our method.
>
> **2. [Re Question 2: The randomness of the proposed method.]**
>
> The proposed method’s randomness arises from two sources: example selection and model prediction.
> - The randomness in the example selection process is due to the independent-cascade diffusion process used to quantify the influence of a specific subset. To ensure stability, we repeat the information diffusion process ten times when quantifying the influence of each potential subset (as illustrated in **Appendix D.2 and Section 2.2 of the revised version draft**). This procedure is a part of our algorithm.
>
> - The randomness in the model prediction process occurs in specific models like GPT-3.5-turbo. When calling GPT-3.5-turbo through OpenAI API and the temperature is not 0, it suffers from randomness.
>
> **3. [Re Question 3: What is the influence of the times of information diffusion?]**
>
> Thank you for your feedback. In our paper, we repeat the information diffusion process ten times when quantifying the influence of each potential subset (**as illustrated in Appendix D.2**). We have moved this content from the Appendix to the main paper (**Section 2.2**).
>
> **4. [Re Question 4: Additional elaboration is required to explain why the combination of IDEAL/vote-k and similarity retrieval proves effective.]**
>
> Thanks for the suggestion. The experimental results suggest that examples with more similar embeddings are more likely to be relevant. Therefore, the combination of IDEAL/vote-k and similarity retrieval is more likely to perform better than its corresponding combination with a random retrieval method.
>
> **5. [Re Question 5: The combination of IDEAL with chain-of-thought prompting.]**
>
>
> Thank you for your constructive feedback. We kindly argue that our method is agnostic to the chain-of-thought technology. Our approach involves selecting a subset of “core” examples to annotate. Once annotated, we can use any prompting technology, including chain-of-thought, to further boost the final performance.
>
>
> **6. [Re Question 6 & 7: Regarding the embedding calculation and corresponding cost.]**
>
>
> Thank you for your suggestions. For ease of analysis, we calculated the embedding using the same settings as previous works Vote-$k$. Specifically, as illustrated in our paper (**page 3 in our submission**), we obtained the embedding of the examples through Sentence-BERT. The cost of this process is negligible and can be ignored, as it takes less than 5 minutes for all datasets in our experiments.
>
>
> **7. [Re Question 8: Regarding the label distribution of the origin data before example selection]**
>
>
> We present the label distribution of the origin data **in the first row of Table 9**. From the statistical results, we can observe that our method achieves the smallest ratio between the numbers of the most frequent class and the least frequent class in 2 out of 3 cases. This scenario implies that our method could mitigate the label skew problem in the origin data.

---

> > ### Comment · Reviewer_RM4K · 2023-11-21
> > **Thanks for responses**
> >
> > Thanks for the authors' responses. My concerns are addressed. I'll keep my score and vote for acceptance.

---

> > > ### Author Response · Authors · 2023-11-22
> > > **To Reviewer RM4K**
> > >
> > > Dear Reviewer RM4K,
> > >
> > > Thank you for taking the time to review our paper. We appreciate your effort and are delighted to receive positive feedback from your comments！
> > >
> > > Best regards,
> > > The Authors

---

### Author Response · Authors · 2023-11-16
**General Response**

We appreciate the reviewers’ insightful comments and constructive feedback on our manuscript. We are pleased to receive positive ratings from most of the reviewers. Furthermore, we are delighted to learn that the reviewers found the research problem to be significant and the core idea to be interesting (Reviewers RM4K, Ra1D, QkNp, and oY6L), the theoretical analysis to be robust (Reviewers RM4K, Ctph, Ra1D), and the experiments to be convincing (Reviewer Ctph, QkNp, oY6L). Based on the reviews, we provide a general response to the points raised by multiple reviewers and individual responses below to address each reviewer’s concerns.

**(1)** Regarding the questions about the experiments, we have taken the following actions:

- For Reviewers Ctph, Ra1D, QkNp, and oY6L, we have either emphasized the location of the required experiments for corresponding comments in our paper or added experiments correspondingly.

- For Reviewer QkNp, we make a justification between our research problem and data leakage problems and we also elaborate on them in more detail in the revised paper.

- For Reviewer oY6L, due to the limited computation resource and rebuttal time, we will try our best to verify our method on all datasets in the ablation sections and repeated the experiments 10 times in the rebuttal time period and update the results accordingly.

- For Reviewers Ctph and oY6L, we have used more content to further explain why some of our experimental results that are not “intuitive”.

- For Reviewer oY6L, we have justified some settings in our experiments. We emphasize that we perform the experiments following the baseline papers.

**(2)** We have addressed the questions about the idea and technical details as follows:

- For Reviewer RM4k, we emphasized the location of some technology details for information diffusions in our paper.

- For Reviewers RM4K and Ra1D, we illustrated the technology details of how to get the embedding and indicated the location for this content in our paper as well.

- For Reviewers Ra1D and QkNp, we further clarified the core motivation/advantage of the proposed influence metric to quantify the selected subset compared with other methods.

- For Reviewers Ctph and oY6L, we illustrated the drawback of Auto-IDEAL in practice, which hindered us from choosing Auto-IDEAL as the main contribution.

**(3)** Missing reference:

- For reviewer Ctph, we have included a detailed discussion of the difference between self-adaptive In-Context learning, TopK, and our work in the related work section in the revised draft.

- For reviewer QkNp, we have included a detailed discussion of "Li et al., Finding Support Examples for In-Context Learning" in our related work section as well.

We also have revised the draft according to all the reviewers. The revisions are highlighted in blue. We sincerely thank all the reviewers for their constructive suggestions. Please feel free to let us know if further details/explanations would be helpful.

Yours truly,
Authors of #816

---

> ### Author Response · Authors · 2023-11-18
> **Further discussion**
>
> Dear Reviewers:
>
> Thanks again for your efforts in reviewing. The discussion deadline is approaching. To enhance this paper, we authors hope that you can check our response and confirm whether there are unclear explanations. We really want to solve them for you.
>
> Yours truly,
> Authors

---

### Meta-Review · Area_Chair_8gtf · 2023-12-05

**Metareview:**

(a) In this paper, the authors propose the method using influence-driven selective annotations to minimize the annotation cost, thereby tackling the high annotation costs of discovering the right prompts for in-context learning. They construct a direct graph to represent unlabeled data and use a diffusion process to quantify the influence of unlabeled subsets.  Some theoretical analysis is done, as is some moderate experiments.
(b)  Gets moderate improvements in results, has some theoretical support, careful analysis of results (clarified in the rebuttal), timing results.
(c)   Results are mild.  More datasets should be tested.  Various improvements acknowledged by the authors need to be done.

**Justification For Why Not Higher Score:**

Wasn't confident about the presentation mode.

**Justification For Why Not Lower Score:**

The reviewer consensus was Accept.

---

### Decision · Program_Chairs · 2024-01-16

Accept (poster)